# GFlowNets Need Automorphism Correction for Unbiased Graph Generation

## Abstract

Generative Flow Networks (GFlowNets) are generative models capable of producing graphs. While GFlowNet theory guarantees that a fully trained model samples from an unnormalized target distribution, computing state transition probabilities remains challenging due to the presence of equivalent actions that lead to the same state. In this paper, we analyze the properties of equivalent actions in the context of graph generation tasks and propose efficient solutions to address this problem. Our theoretical analysis reveals that naive implementations, which ignore equivalent actions, introduce systematic bias in the sampling distribution for both atom-based and fragment-based graph generation. This bias is directly related to the number of symmetries in a graph, a factor that is particularly critical in applications such as drug discovery, where symmetry plays a key role in molecular structure and function. Experimental results demonstrate that a simple reward-scaling technique not only enables the generation of graphs that closely match the target distribution but also facilitates the sampling of diverse and high-reward samples.

## 1 Introduction

Generative Flow Networks (GFlowNets) have emerged as a powerful framework for learning generative models capable of sampling complex, compositional objects with probabilities proportional to a given reward. Inspired by reinforcement learning (RL), GFlowNets generate these objects through a sequence of actions that iteratively modify the structure of the object being built. This approach is particularly well-suited for generating compositional objects, such as graphs, where each step in the process adds components in a structured and interpretable manner. A prominent application of GFlowNets is molecule generation, where molecules are sequentially constructed as graphs (Bengio et al., 2021; Jain et al., 2023a).

However, GFlowNet training objectives rely on the accurate computation of the transition probability $P(s \to s')$ of a policy, which becomes particularly challenging in graph-building environments due to the presence of *equivalent actions*. These are actions that, although different in representation, lead to the same graph structure. For instance, consider Figure 1, where connecting a new node (node 6) to either of two existing nodes (nodes 4 or 5) results in the same graph. Although these actions are distinct, they lead to structurally identical graphs, meaning their transition probabilities must be summed. More generally, when multiple actions lead to the same state $s'$ from a given state $s$, the transition probability must account for all equivalent actions. This issue, referred to as the *equivalent action problem*, arises because determining whether two actions result in the same state requires computationally expensive graph isomorphism tests.

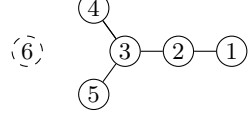

Figure 1: The connected component on the right is the partial graph to be edited, and node 6 is a new node to be connected. Connecting node 6 to any of the existing nodes in the same orbit results in isomorphic graphs.

While GFlowNets were first popularized for their reward-matching capabilities, our analysis reveals that failing to account for equivalent actions introduces a systematic bias in GFlowNets, skewing the model towards sampling graphs with fewer symmetries in atom-based generation and favoring symmetric components in fragment-based generation. This bias is particularly problematic for tasks

such as molecule generation, where symmetry plays a significant role. For example, over 50% of molecules in the ZINC250k dataset exhibit more than one symmetry, with 18% of molecules showing four or more symmetries. Ignoring symmetries leads to incorrect modeling and generation of molecular structures, limiting the diversity and accuracy of samples.

In this paper, we propose a simple yet effective modification to the GFlowNet training objectives to resolve the equivalent action problem. Our method adjusts the reward based on the number of symmetries in a graph, requiring only minimal changes to the existing training algorithms. Additionally, we introduce a new unbiased estimator for the model likelihood, which we use to evaluate the performance of our approach. Our key contributions are as follows:

- We present a rigorous formulation of autoregressive graph generation within the GFlowNet framework, explicitly addressing the equivalent action problem.

- We analyze the impact of equivalent actions on learning and demonstrate how they introduce biases in the sampling process, particularly in tasks involving high-symmetry objects such as molecular graphs.

- We propose a simple yet effective method to resolve the equivalent action problem by scaling the reward based on the automorphism group of the generated graph, allowing GFlowNets to accurately model and sample from the target distribution.

- We introduce an unbiased estimator for the model likelihood, and through theoretical analysis and experiments, demonstrate the effectiveness of our method in generating diverse and high-reward samples.

## 2 RELATED WORK

**Autoregressive graph generation.** There are two primary formulations of autoregressive models: one based on adjacency matrices and the other based on graph sequences (Chen et al., 2021). Methods based on adjacency matrices (You et al., 2018b; Popova et al., 2019; Liao et al., 2019) are unlikely to suffer from the equivalent action problem because they preserve the node order information generated so far, making each pair of (graph, node order) a unique state. In contrast, equivalent actions arise in methods based on graph sequences (You et al., 2018a; Li et al., 2018; Shi et al., 2020). This becomes problematic if a method requires state transition probabilities, as in GFlowNets. Chen et al. (2021) suggest that, for graph sequence-based methods, the size of a node's orbit is equal to the number of equivalent transitions, which inspired our work.

**GFlowNets.** Several learning objectives have been proposed for GFlowNets, including flow matching (Bengio et al., 2021), detailed balance (Bengio et al., 2023), trajectory balance (Malkin et al., 2022), sub-trajectory balance (Madan et al., 2023), as well as their variants to improve training efficiency (Pan et al., 2023; Shen et al., 2023). Recently, GFlowNets have been found to be equivalent to maximum entropy reinforcement learning (Tiapkin et al., 2024; Mohammadpour et al., 2024), which was previously known to be inadequate for directed acyclic graph (DAG) environments (Bengio et al., 2021). However, none of these objectives can avoid the equivalent action problem, as they are formalized based on state transitions, where multiple isomorphic graphs can represent the next state.

Ma et al. (2024) had noticed that equivalent actions must be accounted for to compute exact transition probabilities. They proposed an approximate test to detect equivalent actions at each transition using positional encoding. However, the bias in their model was demonstrated only experimentally on synthetic dataset, without theoretical guarantees. Our work differs in that we provide an exact and efficient solution to this problem, requiring corrections only once at the end of trajectories, as opposed to at each transition within a trajectory, which makes our method straightforward to implement. Our analysis reveals that the bias is present in general setting, namely in both atom- and fragment-based generation schemes, and can significantly impact learning, particularly for highly symmetric graphs. We provide further details in Appendix B and Table 1.

Table 1: Comparison to "Baking Symmetry into GFlowNets"

|  | Ma et al. (2024) | Ours |
|---|---|---|
| Theory | No theoretical guarantees | Theoretical guarantees on biased sampling is provided |
| Method | Approximately identify equivalent actions at each transition | Exactly correct for bias by scaling rewards |
| Types (Generality) | Node types | Node types, edge types, fragments |
| Experiment | Synthetic graphs | Real molecules and synthetic graphs |
| Computation (Exact) | Multiple isomorphism tests for each transition | Computation of $|\mathrm{Aut}(G)|$ once for each trajectory |
| Computation (Approximate) | Multiple positional encoding computations for each transition | Summation over the number of fragments for each trajectory |

## 3 PRELIMINARIES

### 3.1 GRAPH THEORY

Let $G = (V, E)$ denote a graph, where $V = \{v_1, \ldots, v_n\}$ is the set of $n$ vertices, and $E \subseteq V \times V$ is the set of edges. For heterogeneous graphs, we also define labeling functions $l_n$, $l_e$, and $l_g$, which map nodes, edges, and graphs to their respective attributes. We denote $\mathcal{G}$ as the set of all such graphs under consideration. A permutation $\pi$ is a bijective mapping defined on the vertex set. We extend the permutation to the vertex and edge sets as $\pi(V) = \{\pi(v) : v \in V\}$ and $\pi(E) = \{(\pi(v_i), \pi(v_j)) : (v_i, v_j) \in E\}$, as well as to the graph as $\pi(G) = (\pi(V), \pi(E))$. Since any permutation simply relabels node indices, it maps to a structurally identical graph. This notion is formalized as graph isomorphism.

**Definition 3.1** (Isomorphism). *Two graphs $G = (V, E)$ and $G' = (V', E')$ are isomorphic, denoted $G \cong G'$, if there exists a permutation $\pi : V \to V'$ such that $\pi(E) = E'$. For heterogeneous graphs, the permutation must also preserve labels: for every $v \in V$, $l_n(v) = l'_n(\pi(v))$, for every $(u, v) \in E$, $l_e(u, v) = l'_e(\pi(u), \pi(v))$, and $l_g(G) = l'_g(G')$.*

An automorphism is a special case of an isomorphism where the graph is mapped to itself.

**Definition 3.2** (Automorphism). *An automorphism of a graph $G = (V, E)$ is a permutation $\pi$ on the vertex set $V$ that preserves the edge set, meaning $\pi(E) = E$. If labels are present, they must also be preserved under the permutation. The set of all automorphisms of a graph $G$ is called the automorphism group of $G$, denoted by $\mathrm{Aut}(G)$.*

In Figure 1, the graph has two automorphisms: the identity mapping and one that permutes nodes $4$ and $5$. We denote the order (or size) of the automorphism group as $|\mathrm{Aut}(G)|$, which represents the number of symmetries in the graph.

**Definition 3.3** (Orbit). *The orbit of a node $u \in V$ in graph $G$ is defined as $\mathrm{Orb}(G, u) = \{v \in V : \exists \pi \in \mathrm{Aut}(G), \pi(u) = v\}$. Similarly, the orbit of an edge $(u, v) \in E$ in graph $G$ is defined as $\mathrm{Orb}(G, u, v) = \{(h, k) \in E : \exists \pi \in \mathrm{Aut}(G), (\pi(u), \pi(v)) = (h, k)\}$. More generally, the orbit of a node set $S \subseteq V$ in graph $G$ is defined as $\mathrm{Orb}(G, S) = \{S' : \exists \pi \in \mathrm{Aut}(G), \pi(S) = S'\}$.*

An orbit is a set of nodes or edges that are structurally identical. In Figure 1, the orbit of node $4$ is $\{4, 5\}$, and the orbit of the edge $(4, 3)$ is $\{(4, 3), (5, 3)\}$. Equivalent actions occur because they act on nodes in the same orbit; since nodes $4$ and $5$ are in the same orbit, adding a new node to either one is equivalent. This point will be further discussed in Section 4.2.

## 3.2 GENERATIVE FLOW NETWORKS

The generation process of GFlowNets is defined as a finite DAG $(\mathcal{S}, \mathcal{A}, T_{\mathcal{S}})$, where $\mathcal{S}$ and $\mathcal{A}$ are the sets of states and actions, and $T_{\mathcal{S}} : \mathcal{S} \times \mathcal{A} \to \mathcal{S}$ is a deterministic, acyclic transition function. Note that our notation deviates from Bengio et al. (2023), but is similar to (Mohammadpour et al., 2024). This formulation allows two different actions lead to the same next state.

Let $s_0 \in \mathcal{S}$ denote the special starting point of the process, called the initial state, with no incoming edges in the transition graph. Let $\mathcal{X} \subseteq \mathcal{S}$ be the set of terminal states, for which rewards are given. From the initial state $s_0$, objects are constructed sequentially by the forward policy $p_{\mathcal{S}}(a|s)$ until reaching terminal states. A set of complete trajectories, denoted as $\mathcal{T}$, consists of sequences of transitions $\tau = (s_0, a_0, s_1, \ldots, a_{n-1}, s_n)$ starting from the initial state $s_0$ and terminating at $s_n \in \mathcal{X}$, such that $T_{\mathcal{S}}(s_t, a_t) = s_{t+1}$ for all $t$. The goal of GFlowNets is to train a forward policy $p_{\mathcal{S}}$ that generates objects with a probability proportional to their reward, such that $p_{\mathcal{S}}^{\top}(x) = R(x)/Z$, where $Z$ is a normalizing constant and $p_{\mathcal{S}}^{\top}(x)$ denotes the probability of terminating at $x$ when following $p_{\mathcal{S}}$. This is achieved by training $p_{\mathcal{S}}$ using the following objectives.

**Trajectory Balance (Malkin et al., 2022).** The Trajectory Balance (TB) objective is based on the flow consistency constraint at the trajectory level. Given a complete trajectory $\tau$, the TB objective is defined as follows:

$$\mathcal{L}_{\mathrm{TB}}(\tau) = \left( \log \frac{Z \prod_{t=0}^{n-1} p_{\mathcal{S}}(a_t|s_t)}{R(s_n) \prod_{t=0}^{n-1} q_{\mathcal{S}}(s_t, a_t|s_{t+1})} \right)^2 .$$

It introduces a backward policy $q_{\mathcal{S}}$ that reverses the process. Given $q_{\mathcal{S}}$, which can be either fixed or learned, the forward policy $p_{\mathcal{S}}$ and the normalizing constant $Z$ are trained to match the backward flow induced by the reward function and the backward policy.

**Detailed Balance (Bengio et al., 2023).** The Detailed Balance (DB) objective is based on the flow consistency constraint at the state-action level. The objective is defined for each transition $(s, a, T_{\mathcal{S}}(s, a) = s')$ as:

$$\mathcal{L}_{\mathrm{DB}}(s, a, s') = \left( \log \frac{F(s) p_{\mathcal{S}}(a|s)}{F(s') q_{\mathcal{S}}(s, a|s')} \right)^2 .$$

In addition to the backward policy $q_{\mathcal{S}}$, the DB objective requires learning the state flow function $F : \mathcal{S} \to \mathbb{R}^+$, which represents the unnormalized probability that a policy visits state $s$.

## 4 THE EQUIVALENT ACTION PROBLEM

In this section, we formalize the graph generation process in the context of GFlowNets and discuss the equivalent action problem.

### 4.1 PROBLEM DEFINITION

Consider a sequential graph generation process $(\mathcal{G}, \mathcal{E}, T_{\mathcal{G}})$ that constructs graphs by editing the nodes and edges of existing partial graphs, where $\mathcal{E}$ is the set of graph editing actions and $T_{\mathcal{G}} : \mathcal{G} \times \mathcal{E} \to \mathcal{G}$ is an acyclic transition function. In previous work, the relationship between the two processes, $(\mathcal{S}, \mathcal{A}, T_{\mathcal{S}})$ and $(\mathcal{G}, \mathcal{E}, T_{\mathcal{G}})$, was not explicitly addressed, and they were assumed to be identical. Here we relate two processes in a formal way.

Since isomorphism is an equivalence relation, it partitions the space $\mathcal{G}$ into classes, where each graph in a class is structurally identical to the others. Let $[G] = \{G' \in \mathcal{G} : G' \cong G\}$ denote the equivalence class of $G$ induced by graph isomorphism. The state space $\mathcal{S}$ is defined as the set of equivalence classes of graphs, $\mathcal{S} = \{[G] : G \in \mathcal{G}\}$, rather than the graph space $\mathcal{G}$ itself. This is because our goal in using GFlowNets is to sample *any* graph within the equivalence class $s$ in proportion to $R(s)$. If we allow individual graphs to represent states, the equivalence class of a larger graph will be sampled exponentially more often.

In practice, a graph generation process $(\mathcal{G}, \mathcal{E}, T_\mathcal{G})$ is constructed by first designing a set of allowable actions in a given graph. Previous work has defined various types of actions for this process (You et al., 2018a; Li et al., 2018; Bengio et al., 2021; Liao et al., 2019). For example, AddEdge$(u, v)$ adds an edge $(u, v)$ to the existing graph $G$, and AddNode$(u, v)$ adds a new node $v$ and connects it to the existing node $u$. The Stop action can be used to terminate the process, in which case the graph-level attribute is flagged as terminated. We also define the corresponding backward actions that reverse the process. For example, RemoveEdge$(u, v)$ removes the edge $(u, v)$, and RemoveNode$(v)$ removes node $v$ and any edges connected to it. Other types of actions are also possible. By confining the set of forward actions to those that enlarge the graph, the state transitions form a DAG structure.

For a given graph $G$, we define $\text{Orb}(G, \overrightarrow{e})$ as the orbit of an forward graph editing action $\overrightarrow{e} \in \mathcal{E}$, referring to the orbit of the affected node or edge in the graph (See Table 4). For example, the orbit of AddEdge$(u, v)$ is defined as $\text{Orb}(G, u, v)$. Equivalent actions are defined as those that lead to isomorphic graphs within the same orbit.

**Definition 4.1** (Equivalent actions). *Two actions $\overrightarrow{e}'$ and $\overrightarrow{e}'$ are equivalent if they are in the same orbit and induce isomorphic graphs. That is, $\text{Orb}(G, \overrightarrow{e}) = \text{Orb}(G, \overrightarrow{e}')$ and $T_\mathcal{G}(G, \overrightarrow{e}) \cong T_\mathcal{G}(G, \overrightarrow{e}')$. The set of equivalent actions of a graph editing action $\overrightarrow{e}$ given $G$ is denoted as $A(G, \overrightarrow{e})$.*

In Figure 1, adding node 6 to either 4 or 5 results in isomorphic graphs, thus AddNode$(4, 6)$ and AddNode$(5, 6)$ are equivalent actions. Note, however, that the resulting graphs are not equal, as $(4, 6) \neq (5, 6)$. Similarly to the relation between $\mathcal{S}$ and $\mathcal{G}$, the action space $\mathcal{A}$ is defined as the equivalence class of graph editing actions $\mathcal{A} = \{A(G, \overrightarrow{e}) : \overrightarrow{e} \in \mathcal{E}, G \in \mathcal{G}\}$. We denote the backward action corresponding to $\overrightarrow{e}$ as $\overleftarrow{e} = (G, \overrightarrow{e})$. Backward equivalent actions are defined analogously to forward equivalent actions but in the context of the backward graph process.

For computations, we work with graphs rather than directly with states. If we use neural networks equivariant to permutations, such as graph neural networks, then all graph in the same equivalence class will have the same representation. Consequently, we can take one representative graph from the class and treat it as the state, using notations such as $R(s) = R(G)$. However, when defining forward and backward policies over the graph space as $p_\mathcal{G}$ and $q_\mathcal{G}$, it is important to note that $p_\mathcal{G}(\overrightarrow{e}|G) \neq p_\mathcal{S}(a|s)$, as this requires summing over all possible equivalent actions. Given a transition $(G, \overrightarrow{e}, G')$, state-action probabilities can be computed as follows:

$$p_\mathcal{S}(a|s) = \sum_{\overrightarrow{e}' \in A(G, \overrightarrow{e})} p_\mathcal{G}(\overrightarrow{e}'|G) \qquad q_\mathcal{S}(s, a|s') = \sum_{\overleftarrow{e}' \in A(G', \overleftarrow{e})} q_\mathcal{G}(\overleftarrow{e}'|G').$$

Note that by defining equivalent actions as those within the same orbit, we allow for the possibility of two distinct equivalent actions leading to the same next state: $T_\mathcal{S}(s, a) = T_\mathcal{S}(s, a')$ where $a \neq a'$. In rare cases, $p_\mathcal{S}(a|s)$ may not equal the transition probability $P(s \rightarrow s') = \sum_a p_\mathcal{S}(a|s)$. Nevertheless, this definition is sufficient for flow matching because a one-to-one correspondence between $A(G, \overrightarrow{e})$ and $A(G', \overleftarrow{e})$ exists for every transition $(G, \overrightarrow{e}, G')$.

### 4.2 PROPERTIES OF EQUIVALENT ACTIONS

To effectively address the equivalent action problem, we restrict the class of actions we consider to AddNode, AddEdge, SetNodeAttribute, SetEdgeAttribute, SetGraphAttribute, and their corresponding backward actions. For fragment-based graph generation, we also consider AddFragment in our experiments, but we discuss its properties separately in Appendix D. These action classes can be easily extended to cover most of the design space present in previous work on autoregressive graph generation (Li et al., 2018; You et al., 2018a; Luo et al., 2021; Bengio et al., 2021). Precise definitions of these actions are provided in Appendix A.

The definition of the equivalent actions suggests that the computation of state-action probabilities can be simplified by multiplying the number of equivalent actions with a graph editing probability:

$$p_\mathcal{S}(a|s) = |A(G, \overrightarrow{e})| \cdot p_\mathcal{G}(\overrightarrow{e}|G)$$

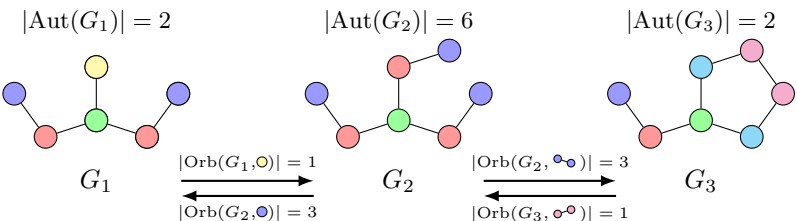

Figure 2: Graphs representing two transitions $(G_1, \overrightarrow{e}_1, G_2, \overrightarrow{e}_2, G_3)$, with the first transition by `AddNode` and the second by `AddEdge`. Graph editing actions induce orbits of some node/edge set, which we used to define equivalent actions. The number of symmetries in the graph is related to the number of equivalent actions, as seen in the ratio $|\mathrm{Aut}(G_t)| : |\mathrm{Aut}(G_{t+1})| = |\mathrm{Orb}(G_t, \overrightarrow{e}_t)| : |\mathrm{Orb}(G_{t+1}, \overleftarrow{e}_t)|$ for $t = 1, 2$. Nodes in the same orbit are given the same color. See Figure 5 for another illustration.

This is because when actions are parameterized using graph neural networks, equivalent actions are assigned equal probabilities. Generally, permutation-equivariant functions, such as graph neural networks, provide the same representations for nodes within the same orbit (more in Appendix F). If we aggregate node representations to obtain edge representations using invariant aggregators, such as SUM or MEAN, the edges in the same orbit will also receive identical representations. This is desired property of graph neural networks, although actions from different orbits may collapse into the same representations, reducing representational power (Zhang et al., 2021). Alternative parameterizations, such as the relative edge parameterization proposed by Shen et al. (2023), also assign equal probabilities to equivalent actions, while potentially enhancing representational power.

Equivalent actions are actions of the same type that act on the same orbit. In other words, graph editing actions operating on the same orbit lead to isomorphic graphs. For instance, if $(u, v)$ and $(h, k)$ are in the same orbit, then `AddEdge`$(u, v)$ and `AddEdge`$(h, k)$ are equivalent actions, meaning they result in isomorphic graphs. This is because only the orbits are structurally important in determining actions.

**Lemma 1.** *For a given graph $G$ and action type, actions applied to the same orbit are equivalent. That is, if $\mathrm{Orb}(G, \overrightarrow{e}) = \mathrm{Orb}(G, \overrightarrow{e}')$, then $T_G(G, \overrightarrow{e}) \cong T_G(G, \overrightarrow{e}')$. For backward actions, $\mathrm{Orb}(G, \overleftarrow{e}) = \mathrm{Orb}(G, \overleftarrow{e}')$ implies $\bar{T}_G(G, \overleftarrow{e}) \cong \bar{T}_G(G, \overleftarrow{e}')$.*

The implication of Lemma 1 is that the number of equivalent actions can be represented as the order of orbits: $|A(G, e)| = |\mathrm{Orb}(G, e)|$. Next, we relate the number of equivalent actions to the order of the automorphism groups.

**Lemma 2.** *Let $G' = T_\mathcal{G}(G, \overrightarrow{e})$, where $\overrightarrow{e} \in \mathcal{E}$ represents an action that either adds a node, an edge, or modifies an attribute in the graph $G$. Then, the following relationship holds:*
$$\frac{|\mathrm{Orb}(G, \overrightarrow{e})|}{|\mathrm{Orb}(G', \overleftarrow{e})|} = \frac{|\mathrm{Aut}(G)|}{|\mathrm{Aut}(G')|}.$$

In Figure 2, we observe that the number of equivalent actions changes as the graph evolves. For instance, from $G_1$, there is only one forward equivalent action, while from $G_2$, there are three. The number of backward actions also varies with each transition, making it seem daunting to account for all equivalent actions step-by-step. However, the ratio of forward equivalent actions to backward equivalent actions between $G$ and $G'$ can be simply expressed as the ratio of the sizes of their automorphism groups. This is the basis for the next theorem.

**Theorem 1** (Automorphism correction)**.** *Let $(G, \overrightarrow{e}, G')$ be a graph transition in atom-based graph generation, and $(s, a, s')$ be a state transition such that $s = [G]$, $a = A(G, \overrightarrow{e})$, and $s' = [G']$. If we use permutation-equivariant functions for $p_\mathcal{G}$ and $q_\mathcal{G}$, then*

$$\frac{p_\mathcal{S}(a|s)}{q_\mathcal{S}(s, a|s')} = \frac{|\mathrm{Aut}(G)|}{|\mathrm{Aut}(G')|} \cdot \frac{p_\mathcal{G}(\overrightarrow{e}|G)}{q_\mathcal{G}(\overleftarrow{e}|G')}.$$

The theorem shows that we can adjust the forward and backward action probability ratio without evaluating all the equivalent actions. The adjustment is determined by the symmetry ratio between two successive states. Given that GFlowNet objectives are based on the ratio of transition probabilities, this adjustment is straightforward to apply, as we will discuss in the next section.

## 5 AUTOMORPHISM-CORRECTED GFLOWNETS (AC-GFN)

In this section, we analyze GFlowNet objectives using our previous results. The following theorem shows that a naive implementation of the TB objective, which does not account for equivalent actions, will train a model biased toward graphs with fewer symmetries.

**Corollary 1** (TB correction). *Assume that $G_0$ is the empty graph or a single node, so that $|\mathrm{Aut}(G_0)| = 1$. Given the complete graph trajectory $\tau = (G_0, \overrightarrow{e}_0, G_1, \ldots, \overrightarrow{e}_{n-1}, G_n)$, the trajectory balance loss can be written as follows:*

$$\mathcal{L}_{\mathrm{TB}}(\tau) = \left( \log \frac{Z \prod_{t=0}^{n-1} p_{\mathcal{G}}(\overrightarrow{e}_t | G_t)}{|\mathrm{Aut}(G_n)| R(G_n) \prod_{t=0}^{n-1} q_{\mathcal{G}}(\overleftarrow{e}_t | G_{t+1})} \right)^2. \tag{1}$$

The equation follows from Theorem 1 and the application of a telescoping sum.

**Implication.** Equation (1) shows that we need to multiply the reward by the order of the automorphism group of the terminal state to properly account for equivalent actions. If we do not scale the reward, we are effectively reducing the rewards for highly symmetric graphs by a factor of $1/|\mathrm{Aut}(G_n)|$. As a result, even if a model is fully trained, the likelihood of reaching the terminal state will not align with the desired distribution; instead, the model is penalized for generating symmetric graphs, following $p_{\mathcal{S}}^{\top}(x) \propto R(x)/|\mathrm{Aut}(G_n)|$. This bias can be easily corrected by evaluating $|\mathrm{Aut}(G_n)|$ and scaling the reward accordingly.

We can similarly adjust the DB objective for each transition, which would require two evaluations of automorphisms for each step. However, we can simply scale the rewards by $|\mathrm{Aut}(G)|$, as in the TB correction, without needing to count automorphisms for each transition. We state this as a theorem and provide a proof in Appendix C.4.

**Theorem 2** (DB correction). *We define the graph-level detailed balance condition, as opposed to the usual state-level condition, as follows:*

$$F(G) p_{\mathcal{G}}(\overrightarrow{e} | G) = F(G') q_{\mathcal{G}}(\overleftarrow{e} | G').$$

*Note that the graph-level detailed balance condition does not account for equivalent actions for each transition. If rewards are given by $\tilde{R}(G) = |\mathrm{Aut}(G)| R(G)$ and the graph-level detailed balance condition is satisfied for all transitions, then the forward policy samples terminal states proportionally to the given reward $R$.*

**Implication.** Together with Corollary 1, we see that scaling the reward alone is sufficient for both TB and DB objectives. This suggests that other GFlowNet objectives, such as subtrajectory balance (Madan et al., 2023), can also be used with reward scaling. This provides a straightforward approach to implementing GFlowNet objectives while reducing the computational burden of counting automorphisms at each transition.

We can interpret the per-transition adjustment for the DB objective as providing intermediate signals for the adjustment, similar to the idea of providing intermediate reward signals, as suggested by Pan et al. (2023). In contrast, reward scaling achieves the same goal by applying the adjustment at the end. See Figure 6 for an illustration of how the graph-level flows can be matched.

Finally, we provide the adjustment formula for fragment-based generation and defer the detailed discussion to Appendix D.

**Theorem 3** (Fragment correction). *Let $G$ represents a terminal state ($[G] \in \mathcal{X}$) generated by connecting $k$ fragments $\{C_1, \ldots, C_k\}$. Then, the scaled rewards to offset the effects of equivalent actions are given by:*

$$\tilde{R}(G) = \frac{|\mathrm{Aut}(G)| R(G)}{\prod_{i=1}^{k} |\mathrm{Aut}(C_i)|} \tag{2}$$

Intuitively, highly symmetric fragments contain many symmetric nodes available for connection, resulting in multiple forward equivalent actions, even though these actions do not lead to distinct outcomes. As a result, without correction, symmetric fragments are more likely to be sampled by the model. Equation (2) corrects this bias by penalizing symmetric fragments.

**Approximate correction method.** Additionally, we experimented with a simplified version where the correction is applied approximately for the fragment-based task. While we can compute exact correction term as in Equation (2), this approximation provides computational benefits, as it avoids counting automorphisms. Moreover, similar approximations can be easily implemented even for more complex generation schemes that do not fit into Equation (2). The approximation works as follows: we assign a number to each fragment based on how many equivalent actions it is likely to incur during generation. We adjust the final rewards by dividing them by the product of the assigned numbers $N$ for the constituent fragments: $R(G)/\prod_{i=1}^{k} N(C_i)$. See more details in Appendix H.

**Computation.** The main additional computation for reward scaling comes from evaluating $|\mathrm{Aut}(G)|$, which is necessary for each trajectory in both the TB and DB objectives. For fragment correction, we can pre-compute $|\mathrm{Aut}(C)|$ in our vocabulary set. While the fastest proven time complexity for computing $|\mathrm{Aut}(G)|$ has remained $\exp(\mathcal{O}(\sqrt{n \log n}))$ for decades (Babai et al., 1983), graphs with bounded degrees can be handled in polynomial time (Luks, 1982). In our experiments, we used the *bliss* algorithm (Junttila & Kaski, 2007), included in the `igraph` package (Csardi & Nepusz, 2006), and did not observe any significant delays in computation. In contrast, removing equivalent actions at each step involves comparing the resulting graphs through graph hashing, which has the same computational cost as graph isomorphism testing. This process requires approximately $K \times T$ more computations compared to our method, where $K$ is the average number of actions per state, and $T$ is the average trajectory length. We provide further analysis on the computation time of counting automorphisms in Appendix G.

**Estimating model likelihood.** As GFlowNets are trained to generate terminal states in proportion to their rewards, one can estimate the maginalized probability $p_{\mathcal{S}}^{\top}(x)$ on a held-out set to compare with $R(x)$. To address the intractability of marginalizing over all trajectories terminating at $x$, Zhang et al. (2022) proposed approximating the model likelihood using importance sampling with $q_{\mathcal{S}}$ as a variational distribution: $p_{\mathcal{S}}^{\top}(x) = \mathbb{E}_{\tau \sim q_{\mathcal{S}}(\tau|x)} \frac{p_{\mathcal{S}}(\tau)}{q_{\mathcal{S}}(\tau|x)}$, where $\tau \in \mathcal{T}$. However, Zhang et al. (2022) worked with a restricted class of decision processes where the equivalent action problem is not present. Instead, we estimate the probability of the terminal state as follows:

$$p_{\mathcal{S}}^{\top}(x) = \mathbb{E}_{\tau \sim q_{\mathcal{G}}(\tau|x)} \left[ \frac{p_{\mathcal{G}}(\tau)}{|\mathrm{Aut}(x)| q_{\mathcal{G}}(\tau|x)} \right] \approx \frac{1}{M|\mathrm{Aut}(x)|} \sum_{i=1}^{M} \frac{p_{\mathcal{G}}(\tau_i)}{q_{\mathcal{G}}(\tau_i|x)}. \tag{3}$$

# 6 EXPERIMENTS

In this section, we conduct experiments to validate our theoretical results and demonstrate the effectiveness of our method. A detailed description of the experiments, including hyperparameters and model used, is provided in Appendix H. We use a uniform backward policy throughout all experiments, and each experiment was run 3 times with different random seeds.

## 6.1 SMALL GRAPHS

We first conduct experiments in a small graph-building environment where graphs are constructed sequentially by adding nodes and edges. We limit maximum number of nodes and edges such that computing the exact model likelihood is tractable ($|\mathcal{X}| = 2,999$). Rewards are assigned based on the number of cycles a graph contains. Since we can compute the exact probabilities of any given state, we compare the probability vectors $p_{\mathcal{S}}^{\top}(x)$ and $R(x)/Z$ for evaluation without approximations. We compare three methods: the TB objective without correction (TB), TB trained on an environment where equivalent actions were removed (TB+RM), and TB with automorphism correction (TB+AC). TB+RM is only feasible for small environments due to the substantial CPU resources required for graph isomorphism tests. However, it provides a baseline that we aim to achieve through reward scaling.

The results are presented in Figure 3. The naive implementation of TB results in limited performance in terms of correlation and $L_1$ errors between probability vectors. In contrast, our method (TB+AC) achieves significantly better performance, reaching similar results of TB+RM, where the equivalent action problem is absent. Upon inspecting the trained normalizing constant, we observed that with

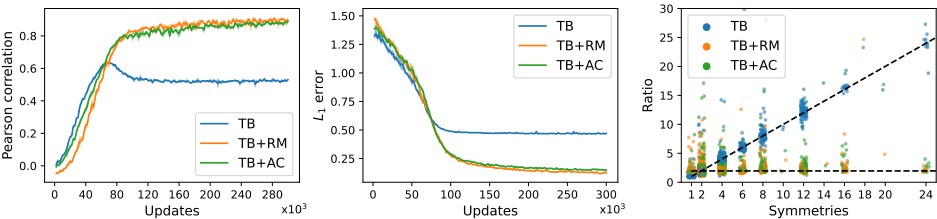

Figure 3: Training results of three methods on the Small Graphs environment. The mean and standard deviation from 3 random seeds are shown in the left two plots, while the rightmost figure is generated from one of the trained models. Symmetries indicate $|\mathrm{Aut}(x)|$, and the ratio represents the target-to-model state probability ratio.

correction, the estimated $Z$ is 6073, closely matching the true value $Z = 6464$. Without correction, however, $Z$ is estimated to be 3277, approximately half of the true value.

The rightmost figure highlights that the results align with our theoretical analysis. The figure shows strip plots grouped by symmetries $|\mathrm{Aut}(x)|$. We define the target-to-model state probability ratio as $\frac{R(x)}{\tilde{Z} p_{\mathcal{S}}^\top(x)}$, where $\tilde{Z}$ is the normalizing constant of the biased target, $\tilde{Z} = \sum_{x \in \mathcal{X}} \frac{R(x)}{|\mathrm{Aut}(x)|}$. Our analysis suggests that the ratio should recover $|\mathrm{Aut}(x)|$ for TB, while it remains constant for unbiased methods. The dashed lines in the plot show our theoretical projection of the ratio for each method. The plot demonstrates that trained sampling distributions match our analysis, revealing the bias in vanilla GFlowNets. Additional results for the DB objective and for uniform targets are shown in Appendix I.

### 6.2 MOLECULE GENERATION

**Task description.** We investigate whether accurately modeling a given target distribution helps generate diverse and high-reward samples in practice. We examine the atom-based generation task from Jain et al. (2023b) and the fragment-based generation task from Bengio et al. (2021). In the atom-based task, the goal is to generate molecules by sequentially adding new atoms, edges, or setting their attributes. Rewards are provided by a proxy model trained on the QM9 dataset, which predicts the HOMO-LUMO gap. In the fragment-based task, we use a predefined set of fragments, each with a predefined set of attachment points—nodes on the fragment where edges can connect. The task involves building a tree graph, where each node represents a fragment, and edges specify the attachment points on the two connected fragments. Rewards are determined by a proxy model that predicts the binding energy of a molecule to the sEH target.

For the atom-based task, we simply scale the final rewards by the order of the automorphism group as described in Equation (1). For the fragment-based task, we additionally correct for fragment automorphisms as described in Equation (2). In GFlowNets, the reward exponent $\beta$ is used to focus sampling on high-reward regions in the state space. The correction is applied after rewards are exponentiated: $C(x)R(x)^\beta$, where $C(x)$ is the correction term.

**Evaluation.** We sampled 5,000 molecules from each method and evaluated the following metrics:

- **Top $K$ diversity.** The average pairwise Tanimoto distance among the top $K$ reward molecules.
- **Top $K$ reward.** The average reward of the top $K$ molecules.
- **Diverse top $K$.** The average reward of the top $K$ molecules, ensuring that each pair has a Tanimoto distance greater than $0.7$.
- **Uniq. fraction.** The fraction of unique molecules in the generated samples.
- **FCS.** Flow Consistency in Sub-graphs (FCS) is the average total variation between the marginal $p_{\mathcal{S}}^\top$ and the target (Silva et al., 2024).

We selected $K = 50$, which corresponds to the top 10% of molecules for our evaluation. When reporting rewards, we adjust them to remove the effects of reward scaling and reward exponents.

Table 2: Results for molecule generation task. Highest scores are highlighted.

| Task | Method | Top $K$ div. | Top $K$ reward | Div. Top $K$ | Uniq. frac. | FCS |
|------|--------|--------------|----------------|--------------|-------------|-----|
| Atom | TB | $0.051_{\pm 0.02}$ | $1.044_{\pm 0.015}$ | $1.044_{\pm 0.015}$ | $0.991_{\pm 0.012}$ | $0.920_{\pm 0.058}$ |
| | TB+AC | $0.073_{\pm 0.03}$ | $1.081_{\pm 0.029}$ | $1.081_{\pm 0.029}$ | $0.999_{\pm 0.001}$ | $0.936_{\pm 0.036}$ |
| Fragment | TB | $0.153_{\pm 0.003}$ | $0.941_{\pm 0.002}$ | $0.941_{\pm 0.002}$ | $1.0_{\pm 0.0}$ | $0.957_{\pm 0.025}$ |
| | TB+XC | $0.164_{\pm 0.008}$ | $0.949_{\pm 0.006}$ | $0.949_{\pm 0.006}$ | $1.0_{\pm 0.0}$ | $0.908_{\pm 0.011}$ |
| | TB+AC | $0.151_{\pm 0.002}$ | $0.952_{\pm 0.003}$ | $0.952_{\pm 0.003}$ | $1.0_{\pm 0.0}$ | $0.975_{\pm 0.017}$ |

**Results.** The results summarized in Table 2 show that accurately modeling the target distribution yields the best results in terms of generating diverse and high-reward samples for both atom-based and fragment-based tasks. This result is noteworthy, given that rewards are negatively correlated with the number of automorphisms in the atom-based task, with a Spearman correlation of -0.36. For the fragment-based task, the Top $K$ diversity remains within the confidence bounds, while the sampled molecules show higher rewards when corrected for automorphisms. We also observe that the approximate correction (TB+XC) already enables the generation of high-reward samples, underscoring the effectiveness and importance of the correction. Without correction, the trained model tends to excessively favor components that incur multiple forward equivalent actions during generation. For example, among 5000 sampled molecules, the vanilla GFlowNet produced 5220 instances of cyclohexane (C1CCCCC1) as its fragments, whereas the corrected method produced only 1042.

In addition, we measured FCS metric and the Pearson correlation between the model's log likelihood $\log p_{\mathcal{S}}^{\top}(x)$ and the log reward $\log R(x)$ on the test set. The purpose of these metrics is to measure the goodness-of-fit of our model to the target distribution. Figure 4 shows that for the atom-based task with a proxy trained on the QM9 dataset, matching the rewards is relatively difficult. This is also evidenced by FCS metric in the Table 2. In contrast, we observe an overall high correlation for the fragment-based task, with further improvements made through corrections.

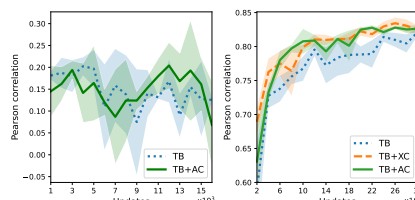

Figure 4: Pearson correlations between rewards and model likelihoods. **Left:** Atom. **Right:** Fragment.

## 7 DISCUSSION AND CONCLUSION

GFlowNets were first proposed as an alternative to previous methods, such as MaxEnt RL (Haarnoja et al., 2017), which are biased toward states with multiple action sequences leading to them in non-injective cases. However, incorrect modeling of state-action probabilities introduces another type of bias in graph generation. While it is unclear how previous works addressed the equivalent action problem, it is likely that they employed the approximation $p_{\mathcal{S}}(a|s) \approx p_{\mathcal{G}}(\vec{e}|G)$. Although we believe that the previous experimental results remain valid if interpreted carefully with the problem in mind, we recommend being explicit about the correction method in all future work.

In this paper, we analyzed the properties of equivalent actions and proposed a simple correction method that allows for unbiased sampling from the target distribution. Our analysis shows that, without correction, highly symmetric graphs are less likely to be sampled, while symmetric fragments are more likely to be sampled, which is crucial for molecule discovery. We demonstrated that the reward scaling technique works for both TB and DB objectives. Experimental results suggest that reward scaling effectively removes bias, allowing for accurate modeling of the target distribution, which is essential for sampling diverse, high-reward molecules. A potential limitation of this paper is that the proposed correction method is demonstrated primarily on specific objectives (TB and DB) and datasets relevant to molecule discovery. Future work could explore applying the method to tasks with different symmetry patterns and reward structures.

## REPRODUCIBILITY

A detailed description of the task, network architecture, and hyperparameters is provided in Appendix H. The proofs of the lemmas and theorems are included in Appendix C. We used an open-source code repository for the molecule generation experiments, which is described in Appendix H. Our code will be made publicly available upon the paper's release.

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

# A   NOTATIONS & DEFINITIONS

In this section, we introduce the key notations and definitions used throughout the paper.

Table 3: Notation

| | | |
|---|---|---|
| | Set of graphs | $\mathcal{G}$ |
| | Set of vertices | $V$ |
| | Set of edges | $E$ |
| | Node labeling function | $l_n$ |
| | Edge labeling function | $l_e$ |
| | Graph labeling function | $l_g$ |
| | Permutation | $\pi$ |
| | Set of automorphisms of graph $G$ | $\text{Aut}(G)$ |
| | Equivalence class of graph $G$ | $[G]$ |
| Graph | Orbit of a node $u$ | $\text{Orb}(G, u)$ |
| | Stabilizer of a node $u$ | $\text{Stab}(G, u)$ |
| | Set of graph editing actions | $\mathcal{E}$ |
| | Graph transition function | $T_{\mathcal{G}}$ |
| | Set of backward graph editing actions | $\bar{\mathcal{E}}$ |
| | Backward graph transition function | $\bar{T}_{\mathcal{G}}$ |
| | Forward policy over graphs | $p_{\mathcal{G}}$ |
| | Backward policy over graphs | $q_{\mathcal{G}}$ |
| | Set of forward equivalent actions | $A(G, \overrightarrow{e})$ |
| | Set of backward equivalent actions | $A(G, \overleftarrow{e})$ |
| | Set of states | $\mathcal{S}$ |
| | Set of actions | $\mathcal{A}$ |
| | State transition function | $T_{\mathcal{S}}$ |
| | Set of terminal states | $\mathcal{X}$ |
| | Reward function | $R$ |
| GFlowNet | Set of complete trajectories | $\mathcal{T}$ |
| | Forward policy over states | $p_{\mathcal{S}}$ |
| | Backward policy over states | $q_{\mathcal{S}}$ |
| | Terminating probability induced by $p_{\mathcal{S}}$ | $p_{\mathcal{S}}^{\top}$ |
| | State flow function | $F$ |

Here we provide the list of actions considered in the paper.

- $\texttt{AddNode}(u, v, t)$ adds a new node $v$ of type $t$ and connects it to node $u$.
- $\texttt{AddEdge}(u, v, t)$ adds a new edge $(u, v)$ of type $t$.
- $\texttt{AddFragment}(C)$ adds a fragment $C$.
- $\texttt{RemoveNode}(v)$ removes node $v$ and all edges connected to it.

- `RemoveEdge`$(u, v)$ removes the edge $(u, v)$.
- `RemoveFragment`$(C)$ removes the subgraph $C$.
- `SetNodeAttribute`$(u, t)$ sets the node-level attribute $t$ for node $u$.
- `SetEdgeAttribute`$(u, v, t)$ sets the edge-level attribute $t$ for the edge $(u, v)$.
- `SetGraphAttribute`$(t)$ sets a graph-level attribute $t$.

The `Stop` action can be considered as setting a terminal flag, and thus, `SetGraphAttribute` can serve as a replacement. Some actions may overlap; for example, rather than allowing `AddNode` to determine the node type, `SetNodeAttribute` could be used instead. It is crucial to design actions in a manner that preserves the DAG structure, ensuring the correct functionality of GFlowNets.

Next, we provide precise definitions of orbits for each action type.

Table 4: Orbit of an action

| | |
|---|---|
| `AddNode`$(u, v, t)$ | $\mathrm{Orb}(G, u)$ |
| `AddEdge`$(u, v, t)$ | $\mathrm{Orb}(G, u, v)$ |
| `AddFragment`$(C)$ | $\mathrm{Orb}(G, V)$ |
| `RemoveNode`$(v)$ | $\mathrm{Orb}(G, v)$ |
| `RemoveEdge`$(u, v)$ | $\mathrm{Orb}(G, u, v)$ |
| `RemoveFragment`$(C)$ | $\mathrm{Orb}(G, C)$ |
| `SetNodeAttribute`$(u, t)$ | $\mathrm{Orb}(G, u)$ |
| `SetEdgeAttribute`$(u, v, t)$ | $\mathrm{Orb}(G, u, v)$ |
| `SetGraphAttribute`$(t)$ | $\mathrm{Orb}(G, V)$ |

## B    COMPARISON TO PRIOR WORK

To the best of our knowledge, Ma et al. (2024) is the only paper addressing the equivalent action problem prior to our work. While the issue of equivalent actions in GFlowNets was identified and partially addressed by Ma et al. (2024), the discussion was limited to experimental validation. In contrast, our work provides the first rigorous theoretical foundation for automorphism correction, demonstrating that this issue is not just experimental but a fundamental and systematic challenge tied to graph symmetries, both for atom-based and fragment-based generation. This finding carries significant implications, especially given that GFlowNets were initially popularized for their reward-matching capabilities. We provide detailed comparison in Table 5.

Table 5: Comparison to "Baking Symmetry into GFlowNets"

| | Ma et al. (2024) | Ours |
|---|---|---|
| Theory | No theoretical guarantees | Theoretical guarantees on biased sampling is provided |
| Method | Approximately identify equivalent actions at each transition | Exactly correct for bias by scaling rewards |
| Types (Generality) | Node types | Node types, edge types, fragments |
| Experiment | Synthetic graphs | Real molecules and synthetic graphs |
| Computation (Exact) | Multiple isomorphism tests for each transition | Computation of $|\mathrm{Aut}(G)|$ once for each trajectory |
| Computation (Approximate) | Multiple positional encoding computations for each transition | Summation over the number of fragments for each trajectory |

# C PROOFS

## C.1 PROOF OF LEMMA 1

We restate Lemma 1.

**Lemma 1** (Restatement of Lemma 1). *For a given graph $G$ and action type, actions applied to the same orbit are equivalent. That is, if $\mathrm{Orb}(G, \overrightarrow{e}) = \mathrm{Orb}(G, \overrightarrow{e}\,')$, then $T_G(G, \overrightarrow{e}) \cong T_G(G, \overrightarrow{e}\,')$. For backward actions, $\mathrm{Orb}(G, \overleftarrow{e}) = \mathrm{Orb}(G, \overleftarrow{e}\,')$ implies $T_G(G, \overleftarrow{e}) \cong T_G(G, \overleftarrow{e}\,')$.*

Our assertion is that graph editing actions of the same type applied to the same orbit are equivalent. This is intuitive for attribute-level actions, and we provide a proof for the `SetNodeAttribute` action.

**Lemma C.1** (SetNodeAttribute). *Let $G[l_n(u) = t]$ denote the graph where the attribute of node $u$ in graph $G$ is changed to $t$. If $\mathrm{Orb}(G, u) = \mathrm{Orb}(G, v)$, then $G[l_n(u) = t] \cong G[l_n(v) = t]$.*

*Proof.* Let us denote the node labeling function of $G[l_n(u) = t]$ as $l_{n[u=t]}$. If $u$ and $v$ are in the same orbit, then by definition, there exists $\pi \in \mathrm{Aut}(G)$ such that $\pi(u) = v$. This permutation $\pi$ satisfies $l_{n[u=t]}(u) = l_{n[v=t]}(v) = l_{n[v=t]}(\pi(u)) = t$, which implies that $\pi$ is an isomorphism between $G[l_n(u) = t]$ and $G[l_n(v) = t]$. $\square$

The proof is nearly identical for the `SetEdgeAttribute` action, with nodes replaced by edges. We now proceed to prove the case for the `AddEdge` action.

**Lemma C.2** (AddEdge). *Let $G[E \cup (u, v)]$ and $G[E \cup (h, k)]$ denote the graphs induced by $E \cup (u, v)$ and $E \cup (h, k)$, respectively. If $(u, v)$ and $(h, k)$ are in the same orbit in $G$, then $G[E \cup (u, v)]$ and $G[E \cup (h, k)]$ are isomorphic. In other words, $\mathrm{Orb}(G, u, v) = \mathrm{Orb}(G, h, k)$ implies $G[E \cup (u, v)] \cong G[E \cup (h, k)]$.*

*Proof.* If $(u, v)$ and $(h, k)$ are in the same orbit, then there exists $\pi \in \mathrm{Aut}(G)$ such that $(\pi(u), \pi(v)) = (h, k)$. Since $\pi$ is an automorphism, it also satisfies $\pi(E) = E$. Thus, $\pi(E \cup (u, v)) = \pi(E) \cup (\pi(u), \pi(v)) = E \cup (h, k)$, indicating that $\pi$ is an isomorphism between $G[E \cup (u, v)]$ and $G[E \cup (h, k)]$. $\square$

**Corollary C.1** (RemoveEdge). *Let $G[E \setminus (u, v)]$ and $G[E \setminus (h, k)]$ denote graphs induced by $E \setminus (u, v)$ and $E \setminus (h, k)$ respectively. Then, $(u, v)$ and $(h, k)$ are in the same orbit in graph $G$ if and only if $G[E \setminus (u, v)]$ and $G[E \setminus (h, k)]$ are isomorphic.*

*Proof.* Let $\tilde{G} = (V, \tilde{E})$ where $\tilde{E} = E \setminus \{(u, v), (h, k)\}$. Then $\tilde{G}[\tilde{E} \cup (u, v)] = G[E \setminus (h, k)]$ and $\tilde{G}[\tilde{E} \cup (h, k)] = G[E \setminus (u, v)]$. Applying Lemma C.2 to the modified graph $\tilde{G}$, we can easily obtain the desired result. $\square$

The proof for `RemoveNode` is provided by Chen et al. (2021) in Appendix 2. `AddNode` can similarly be proved by converting it to `RemoveNode` action.

## C.2 PROOF OF LEMMA 2

We restate Lemma 2 below for completeness.

**Lemma 2** (Restatement of Lemma 2). *Let $G' = T_{\mathcal{G}}(G, \overrightarrow{e})$, where $\overrightarrow{e} \in \mathcal{E}$ represents an action that either adds a node, an edge, or modifies an attribute in the graph $G$. Then, the following relationship holds:*

$$\frac{|\mathrm{Orb}(G, \overrightarrow{e})|}{|\mathrm{Orb}(G', \overleftarrow{e})|} = \frac{|\mathrm{Aut}(G)|}{|\mathrm{Aut}(G')|}.$$

We need the following definition.

**Definition C.1** (Stabilizer). *The stabilizer of a node $u \in V$ in graph $G$ is the set of automorphisms that fix node $u$: $\mathrm{Stab}(G, u) = \{\pi \in \mathrm{Aut}(G) : \pi(u) = u\}$. The stabilizer of an edge $(u, v)$ is defined as $\mathrm{Stab}(G, u, v) = \{\pi \in \mathrm{Aut}(G) : \pi(u) = u, \pi(v) = v\}$. Similarly, the stabilizer of a node set $S$ is defined as $\mathrm{Stab}(G, S) = \{\pi \in \mathrm{Aut}(G) : S = \pi(S)\}$.*

$|\text{Aut}(G_0)| = 1 \qquad |\text{Aut}(G_1)| = 2 \qquad |\text{Aut}(G_2)| = 2 \qquad |\text{Aut}(G_3)| = 6$

Figure 5: Graphs representing three transitions from the initial graph $G_0$. The ratio $|\text{Aut}(G_t)|$ : $|\text{Aut}(G_{t+1})| = |\text{Orb}(G_t, \overrightarrow{e}_t)| : |\text{Orb}(G_{t+1}, \overleftarrow{e}_t)|$ holds for $t = 0, 1, 2$. Nodes in the same orbit are given the same color.

We first prove Lemma 2 for the `AddEdge` action. See Figure 5 for an illustration of the lemma.

**Lemma C.3.** *Let $G = G'[E' \setminus (u, v)]$ and $G' = G[E \cup (u, v)]$ two successive graphs. Then the following equation holds:*

$$\frac{|\text{Orb}(G, u, v)|}{|\text{Orb}(G', u, v)|} = \frac{|\text{Aut}(G)|}{|\text{Aut}(G')|}.$$

*Proof.* Using the orbit-stabilizer theorem, we obtain:

$$\frac{|\text{Orb}(G, u, v)|}{|\text{Orb}(G', u, v)|} = \frac{|\text{Aut}(G)|}{|\text{Aut}(G')|} \iff \frac{|\text{Aut}(G)|}{|\text{Orb}(G, u, v)|} = \frac{|\text{Aut}(G')|}{|\text{Orb}(G', u, v)|}$$

$$\iff |\text{Stab}(G, u, v)| = |\text{Stab}(G', u, v)|,$$

so we can prove the lemma by showing $|\text{Stab}(G, u, v)| = |\text{Stab}(G', u, v)|$. We prove this by showing that $\text{Stab}(G, u, v) = \text{Stab}(G', u, v)$. First, we show that $\text{Stab}(G, u, v) \subseteq \text{Stab}(G', u, v)$. Let $\pi \in \text{Stab}(G, u, v)$. Then, $(\pi(u), \pi(v)) = (u, v)$, and $\pi(E \cup (u, v)) = \pi(E) \cup (\pi(u), \pi(v)) = E \cup (u, v)$, which implies that $\pi \in \text{Stab}(G', u, v)$.

Conversely, let $\pi' \in \text{Stab}(G', u, v)$, which means $(\pi'(u), \pi'(v)) = (u, v)$ and $\pi'(E \cup (u, v)) = \pi'(E) \cup (u, v) = E \cup (u, v)$. For the sake of contradiction, assume $\pi'(E) \neq E$. Then, $\pi'$ must map some element in $E$ to $(u, v)$, which implies $|\pi'(E) \cup (u, v)| \neq |E \cup (u, v)|$ leading to a contradiction. $\square$

For `AddNodeAttribute`$(u, t)$, the proof is simpler. Since the only difference between $G$ and $G'$ is the attribute of node $u$, it is straightforward to see that $\text{Stab}(G, u) = \text{Stab}(G', u)$. The proof for `AddEdgeAttribute` is nearly identical. Next, we prove the same result for the `AddNode`.

**Lemma C.4.** *Let $G' = (V', E')$ be the graph resulted by adding node $v$ to node $u$ in graph $G = (V, E)$. Then,*

$$\frac{|\text{Orb}(G, u)|}{|\text{Orb}(G', v)|} = \frac{|\text{Aut}(G)|}{|\text{Aut}(G')|}.$$

*Proof.* We decompose `AddNode`$(u, v)$ as a sequence of two actions: 1) adding an isolated node $v$ to the connected graph $G$; 2) connecting two nodes $u$ and $v$. Let the intermediate graph be denoted as $G'' = G[V \cup v]$. Using the orbit-stabilizer theorem, we need to show $|\text{Stab}(G, u)| = |\text{Stab}(G', v)|$. First, note that $|\text{Stab}(G, u)| = |\text{Stab}(G'', u)|$, because permutations in both stabilizers fix the node $u$, while isolated node $v$ in $G''$ does not introduce additional symmetry. Then, we only need to show $|\text{Stab}(G'', u)| = |\text{Stab}(G', v)|$, but this is established in Lemma C.3. $\square$

## C.3 PROOF OF THEOREM 1

**Theorem 1** (Restatement of Theorem 1)**.** *Let $(G, \overrightarrow{e}, G')$ be a graph transition in atom-based graph generation, and $(s, a, s')$ be a state transition such that $s = [G]$, $a = A(G, \overrightarrow{e})$, and $s' = [G']$. If we use permutation-equivariant functions for $p_{\mathcal{G}}$ and $q_{\mathcal{G}}$, then*

$$\frac{p_{\mathcal{S}}(a|s)}{q_{\mathcal{S}}(s, a|s')} = \frac{|\text{Aut}(G)|}{|\text{Aut}(G')|} \cdot \frac{p_{\mathcal{G}}(\overrightarrow{e}|G)}{q_{\mathcal{G}}(\overleftarrow{e}|G')}.$$

*Proof.* We can prove the theorem through the following chain of equations using two lemmas in succession:

$$\frac{p_{\mathcal{S}}(a|s)}{q_{\mathcal{S}}(s,a|s')} = \frac{\sum_{\overrightarrow{e}' \in A(G,\overrightarrow{e})} p_{\mathcal{G}}(\overrightarrow{e}'|G)}{\sum_{\overleftarrow{e}' \in A(G',\overleftarrow{e})} q_{\mathcal{G}}(\overleftarrow{e}'|G')}$$

$$= \frac{|A(G,\overrightarrow{e})| \cdot p_{\mathcal{G}}(\overrightarrow{e}|G)}{|A(G',\overleftarrow{e})| \cdot q_{\mathcal{G}}(\overleftarrow{e}|G')}$$

$$= \frac{|\mathrm{Orb}(G,\overrightarrow{e})| \cdot p_{\mathcal{G}}(\overrightarrow{e}|G)}{|\mathrm{Orb}(G',\overleftarrow{e})| \cdot q_{\mathcal{G}}(\overleftarrow{e}|G')}$$

$$= \frac{|\mathrm{Aut}(G)|}{|\mathrm{Aut}(G')|} \cdot \frac{p_{\mathcal{G}}(\overrightarrow{e}|G)}{q_{\mathcal{G}}(\overleftarrow{e}|G')}.$$

$\square$

### C.4 PROOF OF THEOREM 2

Before proving Theorem 2, we first prove the existence of a policy that satisfies graph-level DB constraints.

**Lemma C.5.** *For any given reward function $R$, there exist $p_{\mathcal{G}}$, $q_{\mathcal{G}}$, and $\tilde{F}$ that satisfy the graph-level detailed balance constraints for all transitions $(G, \overrightarrow{e}, G')$, defined as follows:*

$$\tilde{F}(G)p_{\mathcal{G}}(\overrightarrow{e}|G) = \tilde{F}(G')q_{\mathcal{G}}(\overleftarrow{e}|G') \tag{4}$$

*Note that this differs from the usual state-level detailed balance condition:*

$$F(s)p_{\mathcal{S}}(a|s) = F(s')q_{\mathcal{S}}(s,a|s'). \tag{5}$$

*Proof.* By Theorem 1, state-level detailed balance constraints can be rewritten as graph transition probabilities as follows:

$$|\mathrm{Aut}(G)|F(G)p_{\mathcal{G}}(\overrightarrow{e}|G) = |\mathrm{Aut}(G')|F(G')q_{\mathcal{G}}(\overleftarrow{e}|G'). \tag{6}$$

Defining $\tilde{F}(G) = |\mathrm{Aut}(G)|F(G)$, then $\tilde{F}$, $p_{\mathcal{G}}$, and $q_{\mathcal{G}}$ satisfy the graph-level detailed balance constraints for a given $R$. $\square$

**Theorem 2** (Restatement of Theorem 2)**.** *If the rewards are scaled by $|\mathrm{Aut}(G)|$ and the graph-level detailed balance constraints are satisfied for $p_{\mathcal{G}}$, $q_{\mathcal{G}}$, and $\tilde{F}$, then the corresponding forward policy will sample proportionally to the reward.*

*Proof.* For a given complete trajectory $G_0, \overrightarrow{e}_0, \ldots, G_n$, we have:

$$\tilde{F}(G_0)p_{\mathcal{G}}(\overrightarrow{e}_0|G_0) = \tilde{F}(G_1)q_{\mathcal{G}}(\overleftarrow{e}|G_1),$$

$$\cdots$$

$$\tilde{F}(G_{n-1})p_{\mathcal{G}}(\overrightarrow{e}_{n-1}|G_{n-1}) = |\mathrm{Aut}(G_n)|R(G_n)q_{\mathcal{G}}(\overleftarrow{e}_{n-1}|G_n).$$

Multiplying the left- and right-hand sides of all the equations, we get:

$$\tilde{F}(G_0)\prod_{t=0}^{n-1} p_{\mathcal{G}}(\overrightarrow{e}_{t+1}|G_t) = |\mathrm{Aut}(G_n)|R(G_n)\prod_{t=0}^{n-1} q_{\mathcal{G}}(\overleftarrow{e}_t|G_{t+1}).$$

Defining $\tilde{F}(G_0) = Z$, this reduces to the state-level trajectory balance condition with corrections, which ensures $p_{\mathcal{G}}^{\top}(x) \propto R(x)$, as shown by Proposition 1 of Malkin et al. (2022). $\square$

Figure 6: Illustration of the effect of reward adjustment. **Above**: State transitions from and to $s_2$. **Below**: Graph transitions from and to $G_2$. Due to the effect of the scaled reward $\tilde{R}$, state flows are also scaled by $|\text{Aut}(G)|$, leading to $\tilde{F}(G) = F(s)|\text{Aut}(G)|$. The edge flows remain unchanged in this figure. Note that the graph-level detailed balance condition holds, while the termination probability is proportional to $\tilde{R}(s)$.

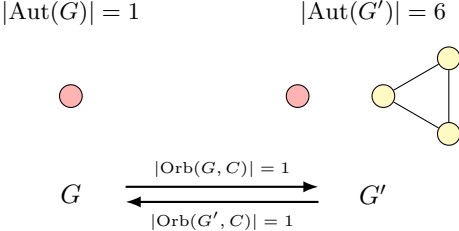

Figure 7: Transition representing the `AddFragment` action.

## D  DISCUSSION ON THE FRAGMENT-BASED GENERATION

In the case of adding a fragment $C$ to the existing graph $G$, resulting in $G' = G \cup C$, we must account for the additional symmetries introduced by the fragment. See Figure 7 for an illustration.

As in atom-based generation, the number of equivalent actions is related to the order of some orbits. We define the orbit of adding a fragment as $\text{Orb}(G, V)$, whose cardinality is 1. This is the number of forward equivalent actions because we only need to choose a fragment from the fragment vocabulary, without considering the existing partial graph. For the backward actions, however, we can remove either $C$ or any subgraph of $G \cup C$ that is isomorphic to $C$. In general, the set of subgraphs of $G \cup C$ that are isomorphic to $C$ under some automorphism in $\text{Aut}(G \cup C)$ are those that, when removed, lead to a graph isomorphic to $G$. This set is precisely the orbit of $C$, denoted as $\text{Orb}(G \cup C, C) = \{V' : \exists \pi \in \text{Aut}(G \cup C), \pi(V_C) = V'\}$.

Next, we can extend Lemma 2 to accommodate the `AddFragment` action, accounting for the symmetries of both the existing graph and the fragment.

**Lemma D.1.** *Let $G = (V_G, E_G)$ be a graph representing the current state. We consider augmenting the graph $G$ by adding a fragment $C = (V_C, E_C)$. Let $G \cup C = (V_G \cup V_C, E_G \cup E_C)$ denote the union of the two graphs (without any edges connecting $G$ and $C$). Then, we have:*

$$\frac{|\text{Orb}(G, V)|}{|\text{Orb}(G \cup C, C)|} = \frac{|\text{Aut}(G)| \cdot |\text{Aut}(C)|}{|\text{Aut}(G \cup C)|}.$$

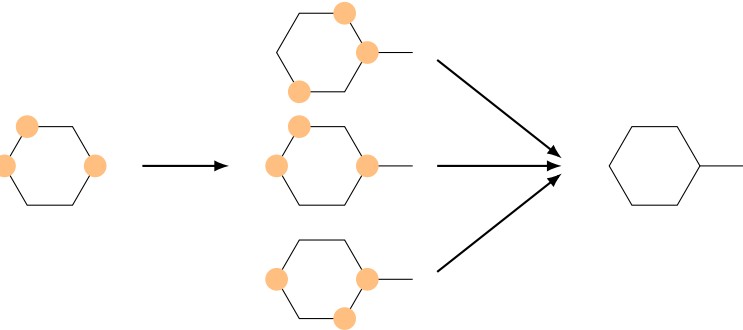

Figure 8: A fragment with attachment points highlighted. Attachment points are designed such that they break symmetries of the fragment. Rightmost graph represent the terminal state where attachment points are removed.

*Proof.* Since $|\mathrm{Orb}(G, V)| = 1$, we only need to consider $|\mathrm{Orb}(G \cup C, C)|$. The stabilizer $\mathrm{Stab}(G \cup C, C)$ is the set of automorphisms in $\mathrm{Aut}(G \cup C)$ that does not mix the labels of $G$ and $C$; it acts independently on $G$ and $C$. Therefore, the order of $\mathrm{Stab}(G \cup C, C)$ is $|\mathrm{Aut}(G)| \cdot |\mathrm{Aut}(C)|$. Using the orbit-stabilizer theorem, we obtain:

$$
\begin{aligned}
|\mathrm{Orb}(G \cup C, C)| &= \frac{|\mathrm{Aut}(G \cup C)|}{|\mathrm{Stab}(G \cup C, C)|} \\
&= \frac{|\mathrm{Aut}(G \cup C)|}{|\mathrm{Aut}(G)| \cdot |\mathrm{Aut}(C)|}.
\end{aligned}
$$

$\square$

Using Lemma D.1, we obtain fragment correction formula in Theorem 3. Unlike atom-based generation, the fragment terms $|\mathrm{Aut}(C)|$ do not cancel out through a telescoping sum. Therefore, these terms must be explicitly accounted for in the correction, both for reward scaling and estimating the model likelihood.

We need to be cautious when applying Lemma D.1, as the exact process of adding fragments may vary depending on the method used. A common approach in the GFlowNet literature is to predefine a set of attachment points for each fragment, which are nodes where an edge can connect to other fragments. Attachment points should be treated as node attributes, even if they are artifacts of the generation process rather than actual molecular properties. This is because they restrict the set of possible actions, including equivalents actions. Thus, even if two nodes, $u$ and $v$, are in the same orbit, they should be considered different if one of them is an attachment point.

These considerations may lead to a situation where we need to define actions like `RemoveAttachmentPoints`, as shown in Figure 8. Conceptually, a graph receives its reward after the attachment points are removed, so that attachment points do not affect the reward. However, this modification introduces three distinct backward actions from the terminal state in the figure, which may complicate the calculation of the backward probabilities. This issue does not arise, however, if we arrange attachment points such that nodes in different orbits (orbits with attachments considered) remain different even after the attachment points are removed. We observe that this holds for the fragments used in Bengio et al. (2021).

## E  RELATION TO NODE ORDERINGS

Some previous work on graph generation uses a distribution over permutations (or node orderings) $\pi$, treating it as a random variable (Li et al., 2018; Chen et al., 2021). Since the node ordering determines the generation order of a graph, the joint probability over the node ordering and state is given by the following:

$$
P(s_n, \pi) = P(s_{0:n}, \pi) = q(\pi | s_{0:n}) p_{\mathcal{S}}(s_{0:n}). \tag{7}
$$

Chen et al. (2021) derived the exact formula for $q(\pi|s_{0:n})$ and trained a model for $p_{\mathcal{S}}(s_{0:n})$. However, the joint probability $P(s_{0:n}, \pi)$ can be easily obtained by multiplying graph-level transition probabilities, without needing to model $p_{\mathcal{S}}(s_{0:n})$ or adjusting for equivalent actions:

$$P(s_{0:n}, \pi) = p_{\mathcal{G}}(G_{0:n}). \tag{8}$$

This result follows because, given a state sequence $s_{0:n}$, the number of different node orderings is equal to the number of possible paths generated by following equivalent actions. In other words, we can interpret different actions that are equivalent as different node orderings that induce the same state sequence. However, previous work on graph generation is not clear about how corrections for equivalent actions were made. This provides a simple formula for computing $P(s_{0:n}, \pi)$ and highlights the importance of distinguishing between graphs and states. See Chen et al. (2021) for more details on using node orderings as a random variable.

## F    PROPERTIES OF GRAPH NEURAL NETWORKS

The key design principle of a graph neural network is permutation equivariance, which ensures that the output remains consistent regardless of how the nodes in the input graph are ordered.

**Definition F.1.** *Let $X \in \mathbb{R}^{n \times d}$ be the node feature matrix of a graph $G$ with $n$ nodes. A matrix-valued function $f$ is permutation equivariant if it satisfies $\pi(f(X, E)) = f(\pi(X), \pi(E))$, where $\pi$ permutes the rows of the matrices.*

Since graph neural networks are permutation equivariant, we can show that they produce identical node representations for nodes in the same orbit.

**Theorem F.1.** *Let $f(X, E)[i]$ represent the $i$-th row of the matrix output by the function $f$. Then, for two nodes $u, v \in V$ in the same orbit, we have $f(X, E)[u] = f(X, E)[v]$.*

*Proof.* Since $u$ and $v$ are in the same orbit, there exists a permutation $\pi$ such that $\pi(X) = X$, $\pi(E) = E$, and $\pi(u) = v$. For this permutation $\pi$, we have:

$$f(X, E)[u] = \pi^{-1} f(\pi X, \pi E)[u]$$
$$= f(\pi X, \pi E)[\pi u]$$
$$= f(X, E)[v],$$

where we omitted brackets for brevity. □

## G    COMPUTATIONAL COST

While computing the exact $|\text{Aut}(G)|$ has inherent complexity, this complexity is unavoidable for exact computation. In practice, fast heuristic algorithms often perform well, particularly for relatively small graphs, and significantly reduce the computational overhead associated with calculating $|\text{Aut}(G)|$. We provide computation time of $|\text{Aut}(G)|$ for several molecular dataset.

Table 6: Computational cost

| Dataset | Sample Size | Num Atoms | Compute time (*bliss*) | Compute time (*nauty*) |
|---------|-------------|-----------|------------------------|------------------------|
| QM9 | 133,885 | 8.8 ± 0.5 | 0.010 ms ± 0.008 | 0.019 ms ± 0.079 |
| ZINC250k | 249,455 | 23.2 ± 4.5 | 0.022 ms ± 0.010 | 0.042 ms ± 0.032 |
| CEP | 29,978 | 27.7 ± 3.4 | 0.025 ms ± 0.014 | 0.050 ms ± 0.076 |
| Large | 304,414 | 140.1 ± 49.4 | - | 0.483 ms ± 12.600 |

Large dataset refers to the largest molecules in PubChem, which is used in the paper Flam-Shepherd et al. (2022). Experiments were conducted on an Apple M1 processor.

Compared to sampling trajectories, which involves multiple forward passes through a neural network, the compute time for $|\text{Aut}(G)|$ is negligible. For comparison, we report the speed of molecular parsing algorithms measured using ZINC250k dataset: 0.06 ms ± 0.70 (SMILES → molecule) and 0.04 ms ± 0.05 (molecule → SMILES). The combination of two parsing steps is often used to check the validity of a given molecule in various prior works. In words, computing $|\text{Aut}(G)|$ is in an order of magnitude faster than validity checking algorithm.

We used the *bliss* algorithm for our experiment. It is easy to use as it is included in the `igraph` package and is fast enough for our purposes. For large molecules, we can still count automorphisms in few milliseconds using the *nauty* package (McKay & Piperno, 2013) as can be seen in the table. We observed that the `pynauty` package does not natively support distinguishing between different edge types, requiring us to transform the input graphs by attaching virtual nodes to handle this. The reported time in the table reflects these preprocessing steps.

While we believe the compute time is already minimal considering current applications, we provide two more recipes to even further improve the run time: 1) Data processing tasks can be easily parallelized across multiple CPUs. Since GFlowNet is an off-policy algorithm, $|\text{Aut}(G)|$ can be computed concurrently with the policy's learning process. 2) For large graphs, fragment-based generation is highly likely to be employed. In such cases, we can utilize an approximate correction formula, as outlined in Appendix D.

## H  EXPERIMENTAL DETAILS

### H.1  SMALL GRAPH

The Small Graphs experiments take place in graph-building environments where homogeneous graphs are constructed edge-by-edge. To enable the exact computation of model likelihood for any terminal state, we restricted the graph size. The total number of states, including the initial empty graph, is 2,300, and the total number of state transitions is 5,734,173.

We used the Adam optimizer (Kingma, 2014) with the default parameters from `PyTorch` (Paszke et al., 2019), setting only the learning rate to 0.0001. We stacked 5 GPS layers (Rampášek et al., 2022). To increase representation power, we augmented node features with one-hot node degree, clustering coefficient, and 4 dimensions of random-walk positional encoding (Dwivedi et al., 2021). The maximum gradient norm was limited to 1 to improve learning stability. For exploration, the policy acted randomly 10% of the time. At each step, 16 samples were collected from the policy, and 48 samples were uniformly drawn from the buffer. We used 64 processors, each with its own buffer of size 1000.

For the DB algorithm, we employed a separate target network to predict backward edge-flows, while the sampling network was trained to match its forward edge-flows to the backward edge-flows. The target network was updated using a moving average with an update rate of 0.99, meaning the target network parameters were incrementally updated by averaging them with the current network parameters at each step. This technique significantly improved the stability of DB training. Performance metrics are computed using the sampling network.

### H.2  MOLECULE GENERATION

We conducted experiments on small molecule generation tasks following Bengio et al. (2021); Jain et al. (2023b). More detailed task descriptions can be found in these previous works. We used a open-source code for tasks.[1] We used a graph transformer architecture (Yun et al., 2019) with the hyperparameters summarized in Table 8 and Table 9.

For the evaluation of Pearson correlation, we used the QM9 test dataset for the atom-based task, while for the fragment-based task, terminal states were sampled by uniformly selecting random actions. The model likelihood was computed using Equation (3) for the atom-based task, with a variant correction term applied for the fragment-based task. We used $M = 5$, and 2048 samples were taken for the test set.

---

[1]https://github.com/recursionpharma/gflownet

Table 7: Hyperparameters for Small Graph experiment

| | Hyperparameters | Values |
|---|---|---|
| | Maximum Nodes | 10 |
| | Maximum Edge | 10 |
| Environment | Maximum Degrees | 4 |
| | Reward Type | 1 + Num. Cycles |
| | Learning Rate ($p_{\mathcal{G}}, Z$) | 0.0001 (Adam) |
| | Batch Size (Online) | 16 |
| | Batch Size (Buffer) | 48 |
| Training | Exploration $\epsilon$ | 0.1 |
| | Gradient Clipping (Norm) | 1.0 |
| | Total Training Steps | 300,000 |
| | Architecture | GPS |
| | Number of Layers | 5 |
| Model | Number of Heads | 4 |
| | Number of Embeddings | 256 |

Table 8: Hyperparameters for atom-based experiments

| | Hyperparameters | Values |
|---|---|---|
| | Learning Rate ($p_{\mathcal{G}}, Z$) | 0.0005 |
| | Batch Size (Online) | 32 |
| | Batch Size (Buffer) | 32 |
| Training | Uniform Exploration $\epsilon$ | 0.1 |
| | Gradient Clipping (Layer-wise Norm) | 10.0 |
| | Reward Exponent $\beta$ | 1 |
| | Number of Updates | 16,000 |
| | Architecture | Graph Transformer |
| | Number of Layers | 4 |
| Model | Number of Heads | 4 |
| | Number of Embeddings | 128 |
| | Number of Final MLP Layers | 1 |

We used the same test samples and model likelihood estimates to compute FCS metrics (Silva et al., 2024). FSC metric is computed as follows:

$$\text{FCS} = \mathbb{E}_{S \sim P_S} \left[ \frac{1}{2} \sum_{x \in S} |p_{\mathcal{S}}^{\top}(x; S) - p(x; S)| \right]$$

where $S$ is a set of sub-graphs sampled from $P_S$, a fully-supported probability distribution over all subsets of $\mathcal{X}$. Marginals $p_{\mathcal{S}}^{\top}$ and the reward distribution $R$ are normalized in the given subset $S$.

### H.3 FRAGMENT CORRECTION METHOD

For fragment-based molecule generation, we used a predefined set of fragments and attachment points provided by Bengio et al. (2021). There are a total of 72 fragments, each with a varying number of attachment points. Our method requires pre-computing the number of automorphisms for each fragment. In Figure 9, we present the number of automorphisms for each fragment used in our experiment. As discussed in Appendix D, attachment points were treated as distinct attributes when counting automorphisms.

Table 9: Hyperparameters for fragment-based experiments

| | Hyperparameters | Values |
|---|---|---|
| | Learning Rate ($p_{\mathcal{G}}$) | 0.0001 |
| | Learning Rate ($Z$) | 0.001 |
| | Batch Size (Online) | 32 |
| | Batch Size (Buffer) | 32 |
| Training | Exploration $\epsilon$ | 0.1 |
| | Gradient Clipping (Layer-wise Norm) | 10.0 |
| | Reward Exponent $\beta$ | 16 |
| | Number of Updates | 30,000 |
| | Architecture | Graph Transformer |
| | Number of Layers | 5 |
| Model | Number of Heads | 4 |
| | Number of Embeddings | 256 |
| | Number of Final MLP Layers | 2 |

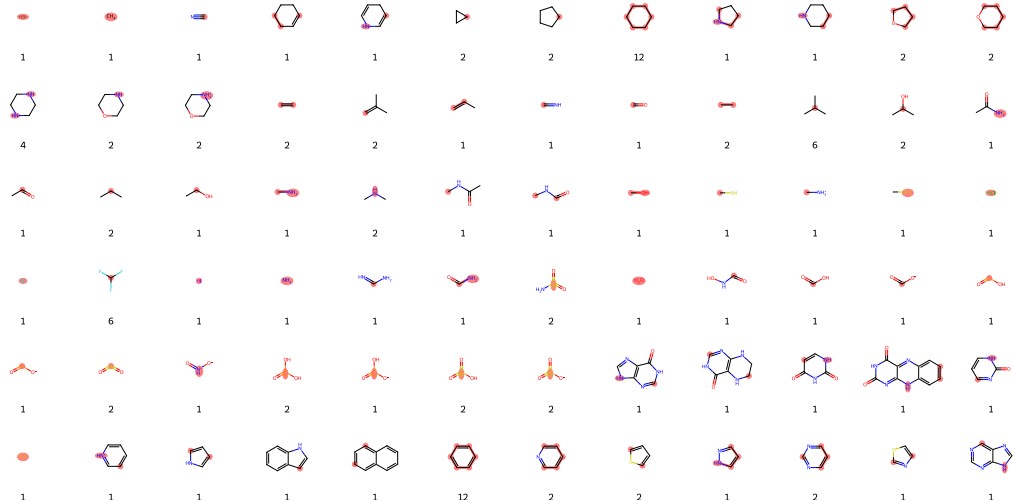

Figure 9: Predefined fragment set used for the fragment-based task. Attachment points, where a single bond can connect to another fragment, are highlighted in red. The numbers indicate the order of the automorphism group of each fragment, $|\mathrm{Aut}(C)|$.

## H.4 APPROXIMATE CORRECTION METHOD

For the fragment-based method, we propose an approximate correction formula, which offers computational benefits. This also aids in understanding the principle behind the correction term for the fragment-based method. The approximation works as follows: we assign a number to each fragment based on how many equivalent actions it is likely to incur during generation. We adjust the final rewards by dividing them by the product of the assigned numbers $N$ for the constituent fragments $C_i$:
$R(X)/\prod_{i=1}^{k} N(C_i)$.

We assigned the number $N$ to each fragment based on how likely it is to incur forward equivalent actions. This is because fragments that incur multiple forward equivalent actions are more likely to be selected if no adjustment is applied. For example, cyclohexane (`C1CCCCC1`) has six attachment points, all in the same orbit, so it will always incur at least six forward equivalent actions in subsequent steps. In contrast, even if a fragment is highly symmetric, if it has only one attachment point, it will incur no equivalent actions. We assigned $N = 1$ to such fragments.

Assuming backward equivalent actions are relatively rare in fragment-based generation, this approximation should closely match the unbiased correction. These numbers were assigned through visual inspection of the fragments. The full set of fragments and their assigned numbers for the approximate correction is provided in Figure 10.

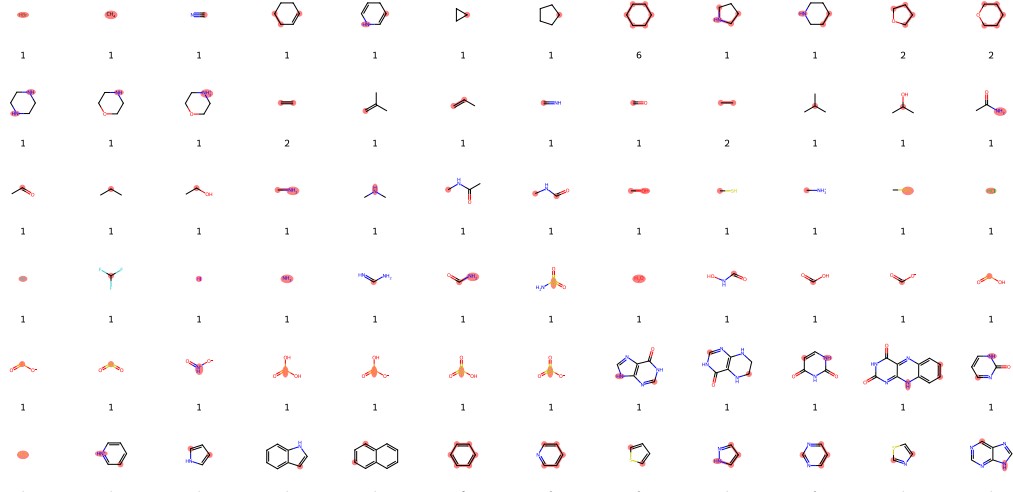

Figure 10: Predefined fragment set used for the fragment-based task. Attachment points are highlighted in red. The numbers below each molecule are used for approximate correction.

# I  ADDITIONAL EXPERIMENTAL RESULTS

## I.1  SMALL GRAPHS

As shown in Theorem 2, we can use both TB and DB objectives with reward scaling. A figure similar to the one in the main text for the DB objective is shown in Figure 11. We observe that removing equivalent actions (DB+RM) improves per-step training efficiency compared to reward scaling (DB+AC), but it comes at a significantly higher computational cost, as discussed in the main text. We utilized 64 processors for DB+RM training, where the wall-clock time became comparable to standard GFlowNet training.

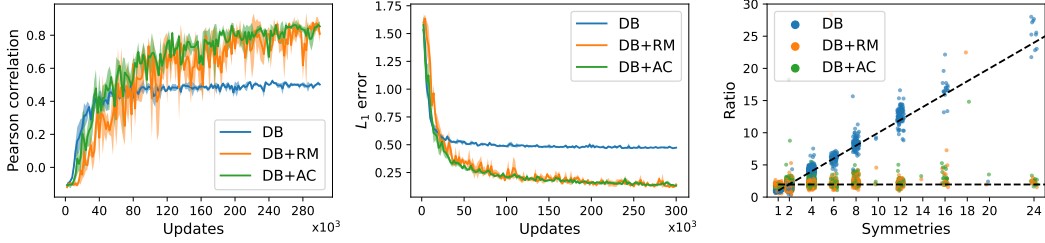

Figure 11: Training results for the DB objective on the Small Graphs environment. The mean and standard deviation from 3 random seeds are shown in the left two plots. The rightmost figure is drawn from one of the trained models.

We observe that reward scaling is effective across several error metrics, including $L_1$, $L_2$, and $L_\infty$, for both DB and TB objectives. See Figure 12 and Figure 13.

Additionally, we conducted experiments with a uniform target distribution, where a reward of 1 is assigned to all terminal states. The optimal model is expected to uniformly sample terminal states, matching $p_S^\top(x) = \frac{1}{Z}$. However, our theoretical analysis reveals that vanilla GFlowNets will be

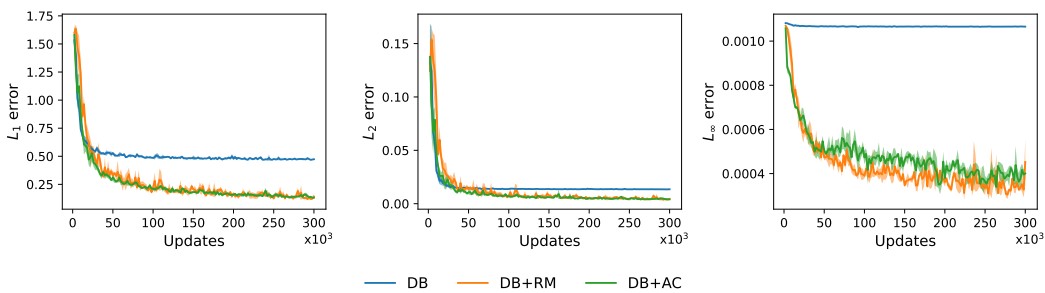

Figure 12: Training results for the DB objective.

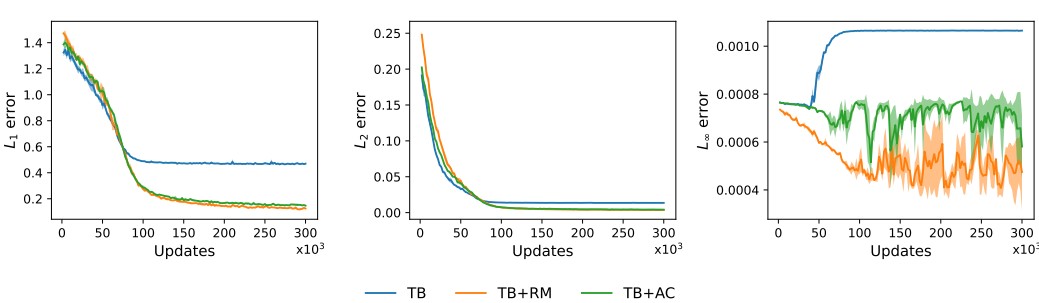

Figure 13: Training results for the TB objective.

trained to match $p_{\mathcal{S}}^{\top}(x) = \frac{1}{\tilde{Z}|\mathrm{Aut}(x)|}$, where $\tilde{Z} = \sum_{x \in \mathcal{X}} \frac{1}{|\mathrm{Aut}(x)|}$. This is demonstrated in Figure 14. As in the main text, the rightmost figure plots $\tilde{Z}/p_{\mathcal{S}}^{\top}(x)$ for all terminal states categorized by the order of automorphisms. This value, referred as target-to-model state probability ratio in the main text, remain constant for optimal model, and recover $|\mathrm{Aut}(x)|$ for biased model. We omit Pearson correlation for this experiment, as uniform target has zero variance.

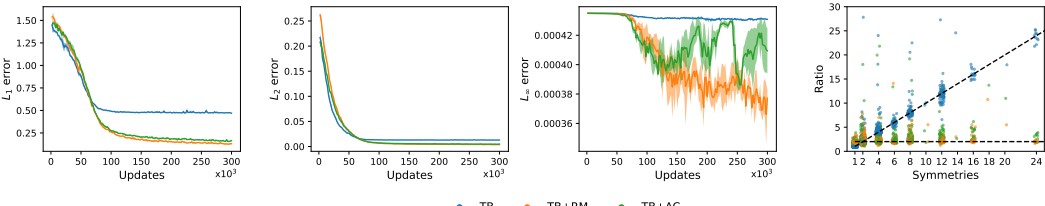

Figure 14: Training results for the TB objective with a uniform target distribution.

## I.2 MOLECULE GENERATION

We investigated which fragments were sampled by each method. We sampled 5,000 terminal states from one of our trained models, resulting in 44,974 fragments used for TB and 44,978 for TB+AC. Symmetric fragments were found to be sampled more frequently in TB, which aligns with our projection, as the fragment correction in TB+AC penalizes symmetric components. However, the proportions of fragments between the two methods are not exactly proportional to the magnitude of the corrections, as some fragments are more likely to occur together (they are not independent).

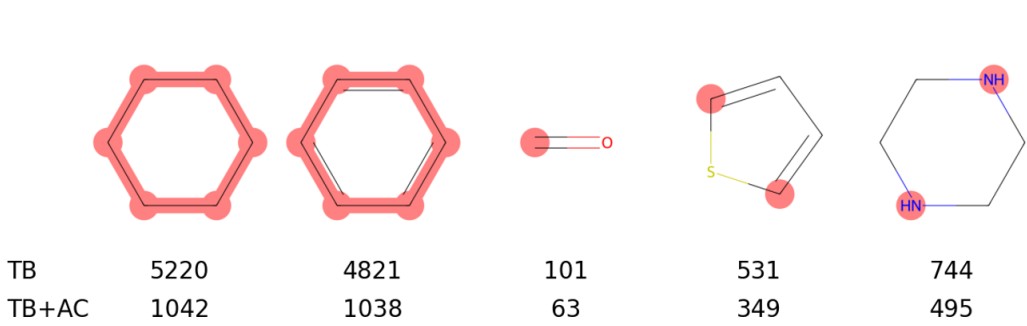

| | | | | | |
|---|---|---|---|---|---|
| TB | 5220 | 4821 | 101 | 531 | 744 |
| TB+AC | 1042 | 1038 | 63 | 349 | 495 |

Figure 15: The number of sampled fragments from 5,000 terminal states for TB and TB+AC. We display the 5 fragments that were sampled most disproportionately.

