# OpenReview forum: "GFlowNets Need Automorphism Correction for Unbiased Graph Generation"
_ICLR.cc/2025/Conference — Submitted to ICLR 2025_

### Official Review · Reviewer_mRgy · 2024-10-26

**Soundness:** 2
**Presentation:** 2
**Contribution:** 2
**Rating:** 5
**Confidence:** 2

**Summary:**

This paper focuses on Generative Flow Networks (GFlowNets), which are generative models used to produce graphs. While GFlowNet theory ensures that a fully trained model can sample from an unnormalized target distribution, the challenge lies in computing state transition probabilities, particularly due to equivalent actions that lead to the same state. The paper analyzes these equivalent actions in graph generation tasks and proposes efficient solutions to mitigate the associated challenges.

**Strengths:**

1. The paper is well organized and the theories are well formulated.

2. The motivation is well introduced.

**Weaknesses:**

1. How does the proposed method compare with other graph generative models, such as flow-based and discrete diffusion-based models?

**Questions:**

N.A.

---

> ### Author Response · Authors · 2024-11-19
>
> ### **Comparison with other graph generative models, such as flow-based and discrete diffusion-based models**
>
> We acknowledge the reviewer's interest in comparing our proposed method with other graph generative models, such as flow-based and discrete diffusion-based models. We provide a brief overview to highlight the distinctions:
>
> **Flow-Based Models:** These models, like normalizing flows, transform simple base distributions into complex ones through a series of invertible and differentiable mappings. This approach allows for exact likelihood computation and efficient sampling, making them effective in continuous data domains such as image generation.
>
> **Discrete Diffusion-Based Models:** These models generate data by learning to reverse a noising process, starting from noise and progressively refining it to match the target data distribution. They have been particularly successful in generating high-quality images and have been extended to other domains.
>
> **GFlowNets:** In contrast, GFlowNets conceptualize generation as a sequential decision-making process, constructing complex objects like graphs through sequences of actions. They aim to sample structures proportional to a given reward function, facilitating the discovery of diverse high-reward structures. This approach is particularly advantageous in scenarios where the target distribution is unnormalized or difficult to sample from directly.
>
> Please note that our work focuses on addressing the unique challenges of equivalent actions in GFlowNets (rather than improving the performances of other graph generative models), improving the robustness and sampling efficiency of GFlowNets in tasks where such challenges are critical bottlenecks, such as graph generation. We believe this distinction is important and will clarify it further in the revision.

---

> ### Author Response · Authors · 2024-12-01
> **Gentle Reminder: Discussion Period Closing Soon**
>
> Dear Reviewer mRgy,
>
> We hope this message finds you well. As the author-reviewer discussion period is set to close in two days, we wanted to kindly remind you of the opportunity to share any additional questions, comments, or suggestions regarding our work.
>
> We would like to highlight our key contributions
>
> - **Rigorous theoretical analysis** of the "equivalent action problem" in GFlowNets, demonstrating that failing to account for equivalent actions introduces a systematic bias, especially in graph generation tasks involving high-symmetry objects.
> - **Novel Correction Method**: Development of an automorphism-based reward-scaling technique to correct the bias, ensuring accurate modeling of the target distribution. This solution applies efficiently to atom- and fragment-based graph generation schemes.
> - **Unbiased Model Likelihood Estimator**: Introduction of an unbiased estimator for model likelihood, allowing for rigorous evaluation of GFlowNet performance in generative tasks.
> - **Efficient Implementation**: Proposed a computationally efficient method for automorphism correction, which requires only one computation per trajectory rather than at each transition, significantly reducing computational overhead.
>
> We sincerely hope that the reviewer takes these contributions into account in their evaluation.
>
> We would deeply appreciate your support in this process and look forward to hearing from you.
>
> Best, Authors

---

### Official Review · Reviewer_6MW9 · 2024-11-01

**Soundness:** 3
**Presentation:** 3
**Contribution:** 3
**Rating:** 6
**Confidence:** 3

**Summary:**

This paper aims to address issues in graph generative GFlowNets that may fail to construct the target distribution due to equivalent actions. Specifically, it analyzes how discrepancies in the number of automorphism groups cause GFlowNets to incorrectly estimate the true reward. To address this, the paper incorporates the number of automorphism groups into the reward function and proves how this corrects reward underestimation. Notably, this paper also considers practical implementation for correction in fragment-based generation. Experimental results show that the proposed method better captures the target distribution.

**Strengths:**

- The paper is well-written and easy to follow, and the proposed method is conceptually straightforward.
- This work is the first to address a significant pitfall in GFlowNets, i.e., errors due to equivalent actions, within primary setting of GFlowNets, i.e., graph generation.
- The authors provide a solid theorem for the corrected objectives showing how their global optima enable GFlowNets to construct the correct target distribution. Although the proofs consider TB and DB, these can also be easily extended to other objectives, such as subTB.
- The experiments are thorough and consider both important settings, namely atom-wise and fragment-wise graph generation.

**Weaknesses:**

No weakness in the major flows. It seems that there are no errors in the proof.

**Questions:**

- To better highlight the pitfalls, I wonder if the authors provide or illustrate a toy-example or toy-experiments where the conventional approaches induce an incorrect generative distribution, e.g., a distribution significantly biased towards the graphs with a low number of automorphism groups.
- Can authors provide the experimental computational costs for computing $|\text{Aut}(s)|$? I am curious about how much overhead the proposed method requires in practice, although the authors provide the time complexity in **Line 378**. Could this overhead be minor relative to time for reward computation or time for sampling trajectories?
- In DB-based implementation, I wonder if there might be improvement in convergence speed when we reparameterize the flow function $F(s)=\tilde{F}|\text{Aut}(s)|$ (like a prior flow reparameterization approach [1]), although this preserves the asymptotic optimality to induce the target distribution.

---

[1] Pan et al., Better Training of GFlowNets with Local Credit and Incomplete Trajectories, ICML 2023

---

> ### Author Response · Authors · 2024-11-19
>
> Thanks for appreciating our work and providing valuable questions!
>
> # Illustrative Example
>
> Thanks for the suggestion! We included the result of the toy experiment for illustrative purposes, but found that it may be difficult to understand without careful reading. In the revised version, we will include a similar experiment with uniform target distribution.
>
> # Computaional Cost
>
> While computing the exact $ |\text{Aut}(s)| $ has inherent complexity, as discussed in the paper, this complexity is unavoidable for exact computation. However, irrespective of the computational cost, fixing the sampling bias due to the ignorance of equivalent actions is a fundamental issue that needs to be resolved. **This correction introduces an inherent computational cost, but it is necessary to maintain the consistency of sampling**. In practice, fast heuristic algorithms often perform well, particularly for relatively smaller graphs, and significantly reduce the computational overhead associated with calculating $ |\text{Aut}(s)| $.
>
> Furthermore, our proposed method requires computing automorphisms only once per trajectory. We present additional experimental results measuring compute time below. Note that the scale of the experiments in our paper corresponds to QM9 and ZINC250k.
>
> ---
> | Dataset  | Sample size | Avg. number of atoms (mean ± std) | Compute time for \|Aut(s)\| using _bliss_ (mean ± std) | Compute time for \|Aut(s)\| using _nauty_ (mean ± std) |
> |-|-|-|-|-|
> | QM9      | 133885      | 8.80 ± 0.51| 0.010 ms ± 0.008  | 0.019 ms ± 0.079|
> | ZINC250k | 249455      | 23.15 ± 4.51| 0.022 ms ± 0.010| 0.042 ms ± 0.032|
> | CEP      | 29978       | 27.66 ± 3.41| 0.025 ms ± 0.014| 0.050 ms ± 0.076|
> | *Large   | 304414      | 140.07 ± 49.38|-| 0.483 ms ± 12.600|
> ---
>
> *Large: the largest molecules in PubChem, data retrieved from https://github.com/danielflamshep/genmoltasks. This data is used in the paper “Language models can learn complex molecular distributions.”
>
> **Experiments were conducted on an Apple M1 processor.
>
> ---
>
> Compared to sampling trajectories, which involves multiple forward passes through a neural network, the compute time for $ |\text{Aut}(s)|$ is negligible. For comparison, we report the speed of molecular parsing algorithms measured using ZINC250k: 0.06 ms ± 0.70 (SMILES → molecule) and 0.04 ms ± 0.05 (molecule → SMILES). The combination of two parsing steps is often used to check the validity of a given molecule in various prior works. In words, computing $ |\text{Aut}(s)|$ is in an order of magnitude faster than validity checking algorithm.
>
> We used the *bliss* algorithm in our paper. It is easy to use as it is included in the igraph package and is fast enough for our purposes. If molecular symmetries grow, such as when symmetric fragments are repeated in polymers, we can still count automorphisms in few milliseconds using the *nauty* package as can be seen in the table. We observed that the pynauty package does not natively support distinguishing between different edge types, requiring us to transform the input graphs by attaching virtual nodes to handle this limitation. The reported time in the table reflects these preprocessing steps.
>
> While we believe the compute time is already minimal considering current applications, we provide two more recipes to even further improve the run time.
>
> - Data processing tasks can be easily parallelized across multiple CPUs. Since GFlowNet is an off-policy algorithm, $ |\text{Aut}(s)|$ can be computed concurrently with the policy's learning process.
> - For large graphs, fragment-based generation is highly likely to be employed. In such cases, we can utilize an approximate correction formula, as outlined in the paper.
>
> In conclusion, the computational overhead of computing automorphism in practice be minor relative to computation of the entire pipeline.
>
> # Effects of Reparameterization
>
> Great question! In our preliminary experiments, we observed no effect of reparameterization on convergence speed. While we are open to further investigation, we offer one possible explanation.
>
> When using a fixed backward policy, we have a unique state flow function, denoted as $F(s)$. This is the function we obtain if corrections are made at every step. On the other hand, if corrections are applied at the end of trajectories, the flow function itself must learn to correct automorphisms. In this case, the flow function becomes $\tilde F(G) = F(s)|\text{Aut}(s)|$ (see figure 6 in the paper).
>
> If we decompose $\tilde F(G)$ into two parts, namely $F(s)$ and $|\text{Aut}(s)|$, it is may happen that the challenging part of the learning process lies in $F(s)$, which is related to reward function $R(s)$. This could explain why intermediate reward signals led to faster learning in previous experiments, whereas intermediate corrections may require further investigation.
>
> We hope this explanation addresses your questions.

---

> > ### Comment · Reviewer_6MW9 · 2024-11-22
> >
> > Thank you for the detailed response. However, I would like to defer the final assessment until the authors provide a detailed comparison with "Baking Symmetry into GFlowNets."

---

> > > ### Comment · Reviewer_6MW9 · 2024-11-25
> > >
> > > After reviewing the above clarification and existing work "Baking Symmetry into GFlowNets", I believe this work loses novelty in its motivation but still has a basic value to be accepted due to (1) technical improvements, (2) extensive evaluation, and (3) theoretical contributions. Therefore, my score is 6 at this time.

---

### Official Review · Reviewer_Rf27 · 2024-11-03

**Soundness:** 3
**Presentation:** 3
**Contribution:** 3
**Rating:** 6
**Confidence:** 3

**Summary:**

This work first studies the properties of equivalent actions when applying GFlowNets for graph generation. Equivalent actions denote the set of actions that lead to isomorphic graphs at each step of the autoregressive generation process. This work provides a theoretical analysis on the impact of ignoring equivalent actions and points out that it would introduce bias in the sampling distribution. With this insight, this work further proposes a simple correction on the GFlowNet objectives by using the order of the automorphism group to account for equivalent actions. This can correct the reward for highly symmetric graphs.

Experiments on small graph generation and small molecule generation are conducted to show the performance of the proposed correction.

**Strengths:**

(1) It is really interesting and valuable to the community to identify the impact of ignoring equivalent actions in GFlowNets for graph generation. The theoretical analysis is quite sound from my reading and I think it is valuable to other readers.

(2) The theoretical results are quite elegant, thus leading to a simple correction to the original GFlowNets objectives. It is quite enjoyable to see that the correction term is the order of the automorphism group.

(3) The experiments can show that with such a simple correlation, the sampling bias and resulting performance are notably improved, which can support the theoretical analysis and the proposed corrected objectives straightforwardly.

(4) The paper is well written.

**Weaknesses:**

(1) It looks really computationally expensive to evaluate the order of the automorphism group and the complexity could increase exponentially with the size of the graph. I understand that the paper provides some analysis on the computation. However, the experimental study on the complexity is missing, while it is very important to assess the practical usefulness of the proposed idea.

(2) I am a bit concerned about the practicality of the method. The experiments are mainly on small graph and small molecule generation. It is unclear if this method can be scalable to generate large molecules.

**Questions:**

See the weaknesses section

---

> ### Author Response · Authors · 2024-11-19
>
> Thanks for appreciating our work and providing a great summary!
>
> # Computational Cost
>
> While computing the exact $ |\text{Aut}(s)| $ has inherent complexity, as discussed in the paper, this complexity is unavoidable for exact computation. However, irrespective of the computational cost, fixing the sampling bias due to the ignorance of equivalent actions is a fundamental issue that needs to be resolved. **This correction introduces an inherent computational cost, but it is necessary to maintain the consistency of sampling**. In practice, fast heuristic algorithms often perform well, particularly for relatively smaller graphs, and significantly reduce the computational overhead associated with calculating $ |\text{Aut}(s)| $.
>
> Furthermore, our proposed method requires computing automorphisms only once per trajectory. We present additional experimental results measuring compute time below. Note that the scale of the experiments in our paper corresponds to QM9 and ZINC250k.
>
> ---
> | Dataset  | Sample size | Avg. number of atoms (mean ± std) | Compute time for \|Aut(s)\| using _bliss_ (mean ± std) | Compute time for \|Aut(s)\| using _nauty_ (mean ± std) |
> |-|-|-|-|-|
> | QM9      | 133885      | 8.80 ± 0.51| 0.010 ms ± 0.008| 0.019 ms ± 0.079|
> | ZINC250k | 249455      | 23.15 ± 4.51| 0.022 ms ± 0.010| 0.042 ms ± 0.032|
> | CEP      | 29978       | 27.66 ± 3.41| 0.025 ms ± 0.014| 0.050 ms ± 0.076|
> | *Large   | 304414      | 140.07 ± 49.38| -| 0.483 ms ± 12.600|
> ---
>
> *Large: the largest molecules in PubChem, data retrieved from https://github.com/danielflamshep/genmoltasks. This data is used in the paper “Language models can learn complex molecular distributions.”
>
> **Experiments were conducted on an Apple M1 processor.
>
> ---
>
> Compared to sampling trajectories, which involves multiple forward passes through a neural network, the compute time for $ |\text{Aut}(s)|$ is negligible. For comparison, we report the speed of molecular parsing algorithms measured using ZINC250k: 0.06 ms ± 0.70 (SMILES → molecule) and 0.04 ms ± 0.05 (molecule → SMILES). The combination of two parsing steps is often used to check the validity of a given molecule in various prior works. In words, computing $ |\text{Aut}(s)|$ is in an order of magnitude faster than validity checking algorithm.
>
> Our primary focus in this work is on **small molecule generation for drug discovery**, where smaller molecular sizes are most relevant. These sizes align with the practical requirements of many real-world drug discovery tasks, making our experiments and methodology well-suited to this domain.
>
> That said, we emphasize that our method is not inherently limited to small molecules and can extend to larger molecules. The scalability of the approach depends on the computational efficiency of the symmetry calculations, and modern graph-processing tools enable handling larger molecular structures effectively. While the specific experiments in our paper focus on small molecules, the underlying principles and methodology remain applicable to larger graphs, provided appropriate computational resources and preprocessing techniques are employed. Furthermore, the compute time for counting automorphisms for large molecules is as small as few milliseconds as reported in the table.
>
> While we believe the compute time is already minimal considering current applications, we provide two more recipes to even further improve the run time.
>
> - Data processing tasks can be easily parallelized across multiple CPUs. Since GFlowNet is an off-policy algorithm, $\text{|Aut(s)|}$ can be computed concurrently with the policy's learning process.
> - For large graphs, fragment-based generation is highly likely to be employed. In such cases, we can utilize an approximate correction formula, as outlined in the paper.
>
> In conclusion, the computational overhead of computing automorphism in practice is minor relative to computation of the entire pipeline.
>
> We hope this addresses your concerns.

---

> > ### Comment · Reviewer_Rf27 · 2024-11-22
> >
> > Thanks for the clarification.
> >
> > Could you provide the running time comparison between w/ the proposed correction and w/o the proposed correction? It's hard to show that the time overhead is minor from these numbers in the table. What really matters is the relative time overhead.

---

> > > ### Author Response · Authors · 2024-11-24
> > >
> > > Thanks for giving us the opportunity to further clarify our method.
> > >
> > > The table below summarizes the runtime for different configurations of atom-based and fragment-based generation methods. To ensure fairness, we re-run all experiments to disable parallel computation, using single processor and GPU. Training steps were limited to 1,000, with all other settings kept consistent with the original paper. We report (mean ± std) with three runs.
> > >
> > > ---
> > > |  | ***Atom** | ****Fragment** |
> > > | --- | --- | --- |
> > > | No corrections (Vanilla GFlowNet) | 44.60 min ± 4.69 | 23.84 min ± 0.45 |
> > > | Exact reward scaling (**ours**) | 49.47 min ± 3.14 | 27.17 min ± 2.62 |
> > > | Approximate reward scaling (**ours**) | - | 24.92 min ± 2.96 |
> > > | Exact isomorphism tests (Ma et al. [1]) | 276.96 min ± 6.28 | 385.12 min ± 12.12 |
> > > ---
> > >
> > > ***Atom:** atom-based generation, with rewards given by a proxy trained on QM9 dataset.
> > >
> > > ****Fragment:** fragment-based generation, with rewards given by a proxy that predicts binding energy to the sEH target.
> > >
> > > [1] Ma et al., “Baking Symmetry into GFlowNets,” 2024.
> > >
> > > ---
> > >
> > > When no corrections are applied, the fragment-based method is faster due to its shorter trajectories. However, when exact isomorphism tests are introduced, the computational cost increases significantly. Specifically, the fragment-based method with exact isomorphism tests incurs the highest computational cost (385 min), reflecting the impact of handling larger molecules.
> > >
> > > On the other hand, our method introduces minimal additional overhead, making it a practical alternative for both atom-based and fragment-based generation tasks, as the differences in runtime are within the standard deviations. Additionally, we used open-source code for the experiments, making only minor changes to the original implementation. Consequently, there is some additional overhead due to the conversion of data types. We believe this overhead could be eliminated if our method were seamlessly integrated into the pipeline.
> > >
> > > We hope this addresses your concerns.

---

### Official Review · Reviewer_6mnv · 2024-11-04

**Soundness:** 3
**Presentation:** 3
**Contribution:** 3
**Rating:** 5
**Confidence:** 4

**Summary:**

This work points out that automorphic actions may cause GFlowNets to artificially over/undersample terminal states compared to the target distribution. They also propose a fix by re-scaling the reward to account for the size of the automorphism group of terminal states. They illustrate the pathology and their fix in a toy example and a molecule generation task.

**Strengths:**

* This work is overall well-written and easy to follow;
* It shows that a common practice of treating graphs as if they were the GFlowNet states leads to incorrect sampling --- implying that, to some extent, there is a series of incorrect experiments in the GFlowNet literature;
* Authors provide a quick fix to the issue.

**Weaknesses:**

* It appears Figure 3 uses Equation 3 to compute the final state probabilities. I am not sure this is a fair evaluation. I suggest the authors use the empirical approximations of the distributions over terminal states (based on GFlowNet samples) for comparison. For instance, measuring L1 between the empirical sampling distribution and the target.

* The metrics in Table 1 have no direct relationship to goodness-of-fit. I understand enumerating the terminal states is impossible for extensive supports, making computing the L1 distance to the target unfeasible. Nonetheless, authors could use the FCS [1] as a proxy. Otherwise, we cannot draw conclusions about sampling correctness in large environments.

* Authors said the additional cost of running BLISS in the final states is negligible. I reckon this should be task-specific. This shouldn't intuitively be negligible if all intermediate states are also final. Please elaborate on the discussion and provide numbers/experimental evaluations.

* The experimental campaign is relatively short compared to recent works on GFlowNets.

* While I value the authors' contribution, I believe their contributions and derivations are somewhat straightforward and the work's novelty is limited.

[1] https://openreview.net/forum?id=B8KXmXFiFj

**Questions:**

* It would be nice to see an illustration of the bias authors point to using a uniform target. Then, plotting the marginal over the size of automorphism relations for each sample should highlight this bias.

---

> ### Author Response · Authors · 2024-11-19
>
> Thanks for your valuable feedback and suggestions!
>
> # Evaluation
> ### **On Evaluation**
>
> To clarify, in Figure 3 (the results of the toy experiment), we computed state probabilities by enumerating all possible trajectories, rather than using Equation 3. This approach was feasible because we constrained the problem size, making the exact computation of state probabilities tractable. Additionally, we took advantage of the many overlapping sub-trajectories, which allowed us to eliminate redundant computations.
>
> ### **On goodness-of-fit, diversity, and rewards**
>
> The purpose of the metrics presented in Table 1 is to show the effects of the correction on downstream tasks, specifically in discovering diverse molecules and high-reward molecules, rather than to demonstrate the goodness-of-fit of the proposed method. To assess the goodness-of-fit, we measured Pearson correlation between $\log R(x)$ and $\log P_F(x)$ as shown in Figure 4. Estimating $\log P_F(x)$ was possible without enumeration using empirical approximation we proposed in Equation 3. Although we understand that Pearson correlation is not a perfect measure as it is scale-invariant, it is considered as a good proxy in practice. While we appreciate your suggestion regarding the FCS metric, we are not entirely certain about its implementation, as the code is unavailable. Since the FCS metric requires estimating the marginal distribution over terminal states, we believe it could be used with our proposed estimation method in the future. We would be more than happy to use FCS if once the code becomes available.
>
> # Experiments
>
> Unlike other papers on GFlowNets, the purpose of our paper is not to simply improve the performance of GFlowNets. Rather, the work is focused on identifying the critical bias present in previous work of GFlowNets. That said, we are willing to include more experimental results in revised paper, including illustrative experiments using uniform target.
>
> We excluded the result of uniform target from the paper because the Pearson correlation cannot be computed (as the uniform target has zero variance). However, we are happy to include it in the revised version. The plots are very similar to Figure 3, however, the result based on the number of rings.
>
> # Technical contribution
>
> We emphasize that seemingly simple findings often have the potential to be highly impactful. We would like to highlight that our work is the first work to theoretically address the "equivalent action problem," which is particularly critical when using GFlowNets to model target distributions. This problem, previously overlooked, fundamentally affects the correctness of the sampling distribution in GFlowNets.
>
> Beyond correcting this issue, our findings propose a new method to estimate model likelihood, which has significant implications for various graph-related tasks. Moreover, our formulation, which explicitly distinguishes between states and graphs, provides a fresh perspective that we believe can offer valuable insights into other graph-centric applications.
>
> Additionally, as discussed in Appendix D, we briefly outline how our findings can be extended to improve methods that incorporate "node ordering" as a variable. This demonstrates that the implications of our work extend beyond GFlowNets and can influence a broader range of methodologies. We hope these contributions underscore the novelty and potential impact of our research.

---

> ### Author Response · Authors · 2024-11-19
>
> # Computational Cost
>
> While computing the exact $|\text{Aut}(s)| $ has inherent complexity, as discussed in the paper, this complexity is unavoidable for exact computation. However, irrespective of the computational cost, fixing the sampling bias due to the ignorance of equivalent actions is a fundamental issue that needs to be resolved. **This correction introduces an inherent computational cost, but it is necessary to maintain the consistency of sampling**. In practice, fast heuristic algorithms often perform well, particularly for relatively smaller graphs, and significantly reduce the computational overhead associated with calculating $ |\text{Aut}(s)| $.
>
> Furthermore, when compared to the cost of sampling trajectories, which involves multiple forward passes through a neural network, the compute time for $|\text{Aut}(s)|$ remains still negligible. Also it is important to note that our proposed method requires computing automorphisms only once per trajectory. To address your comment, we provide additional experimental results measuring the compute, as shown below. Note that the scale of the experiments in our paper corresponds to QM9 and ZINC250k.
>
> ---
> | Dataset  | Sample size | Avg. number of atoms (mean ± std) | Compute time for \|Aut(s)\| using _bliss_ (mean ± std) | Compute time for \|Aut(s)\| using _nauty_ (mean ± std) |
> |-|-|-|-|-|
> | QM9| 133885| 8.80 ± 0.51| 0.010 ms ± 0.008  | 0.019 ms ± 0.079|
> | ZINC250k | 249455| 23.15 ± 4.51| 0.022 ms ± 0.010| 0.042 ms ± 0.032|
> | CEP  | 29978| 27.66 ± 3.41| 0.025 ms ± 0.014| 0.050 ms ± 0.076|
> | *Large | 304414| 140.07 ± 49.38|-| 0.483 ms ± 12.600|
> ---
>
> *Large: the largest molecules in PubChem, data retrieved from https://github.com/danielflamshep/genmoltasks. This data is used in the paper “Language models can learn complex molecular distributions.”
>
> **Experiments were conducted on an Apple M1 processor.
>
> ---
>
> For comparison, we report the speed of molecular parsing algorithms measured using ZINC250k: 0.06 ms ± 0.70 (SMILES → molecule) and 0.04 ms ± 0.05 (molecule → SMILES). The combination of two parsing steps is often used to check the validity of a given molecule in various prior works. In words, computing $ |\text{Aut}(s)|$ is in an order of magnitude faster than validity checking algorithm. Even if we compute automorphisms for all intermediate states, this amounts to less than 20x increase for small molecules, which is less than a millisecond.
>
> We used the *bliss* algorithm in our paper. It is easy to use as it is included in the igraph package and is fast enough for our purposes. For large molecules, we can still count automorphisms in few milliseconds using the *nauty* package as can be seen in the table. We observed that the pynauty package does not natively support distinguishing between different edge types, requiring us to transform the input graphs by attaching virtual nodes to handle this limitation. The reported time in the table reflects these preprocessing steps.
>
> While we believe the compute time is already minimal considering current applications, we provide two more recipes to even further improve the run time.
>
> - Data processing tasks can be easily parallelized across multiple CPUs. Since GFlowNet is an off-policy algorithm, $ |\text{Aut}(s)|$ can be computed concurrently with the policy's learning process.
> - For large graphs, fragment-based generation is highly likely to be employed. In such cases, we can utilize an approximate correction formula, as outlined in the paper.
>
> In conclusion, the computational overhead of computing automorphism in practice be minor relative to computation of the entire pipeline.

---

> > ### Comment · Reviewer_6mnv · 2024-11-25
> >
> > I have read the authors' rebuttal and other reviewers' comments, as well as Emmanuel's public comment. I believe this work loses significant novelty in light of "baking symmetry into GFlowNets." Some of my concerns were not sufficiently addressed, e.g., using a proper correctness measure for large supports and promoting a comparison using empirical distributions directly (instead of Eq. 3), among others. Therefore, I am keeping my score.

---

> > > ### Author Response · Authors · 2024-12-01
> > >
> > > Thank you for your thoughtful feedback and for taking the time to engage with our rebuttal. We respect your comments and would like to address your remaining concerns in more detail.
> > >
> > > 1. **Novelty and Contributions:**
> > >
> > >     We believe our paper presents at least three key novelties, each with significant implications for future research:
> > >
> > >     - **Rigorous theoretical analysis:** We have identified the exact bias present in GFlowNet training. While the problem was previously noted, our work is the first to formulate the problem, theoretically justifying the motivation for employing the correction.
> > >     - **Novel Correction Method**: We proposed a practical solution to address this bias, making unbiased GFlowNet training feasible in practice.
> > >     - **Unbiased Model Likelihood Estimator**: We introduced a novel model likelihood estimator, which serves as a fundamental measure for evaluating generative models.
> > >
> > >
> > > 2. **Correctness Measures:**
> > >
> > >     In the revised version, we included FCS [1] as an evaluation metric to measure model correctness. We agree that FCS is an intuitive and appropriate alternative for measuring correctness, but it should be used in combination with our model likelihood estimator. As such, we plan to further explore the aspects of FCS when used with our proposed estimator, as it has not been previously tested for graph generation tasks.
> > >
> > >     [1] https://openreview.net/forum?id=B8KXmXFiFj
> > >
> > >
> > >
> > > 3. **Empirical Distributions and Model Evaluation:**
> > >
> > >     For synthetic graphs, exact marginal state probabilities can be computed directly, which eliminates the need for either Eq. 3 or empirical distributions for evaluation. However, we understand your concerns about using Eq. 3 for model evaluation. To address this, we will include additional experimental results comparing different model likelihood estimators in conjunction with the FCS metric in a future version of the paper.
> > >
> > > We hope this clarifies our contributions and provides further insights into our work. We would greatly appreciate any reconsideration of the assigned score.
> > >
> > > Regards,
> > >
> > > Authors

---

### Official Review · Reviewer_ynfw · 2024-11-04

**Soundness:** 3
**Presentation:** 3
**Contribution:** 3
**Rating:** 6
**Confidence:** 4

**Summary:**

This paper shows that the so-called problem of equivalent actions (appearing from graph symmetries) biases graph generation processes in GFlowNets. To tackle this, the paper proposes a simple correction procedure that scales the reward function by the number of symmetries of the associated graph (terminal state).  Experiments on artificial data and molecule generation tasks aim to show the effectiveness of the proposed approach.

**Strengths:**

- **Clarity**: Overall, the text is well-written and easy to follow (despite some overloaded notation);
- **Motivation and Relevance**: The motivation is clear, and the problem is relevant as graph (molecule) generation is one of the main applications of GFlowNets;
- **Flexibility**: The proposed correction procedure is flexible as it applies to different training schemes (balance conditions).

**Weaknesses:**

- **Computational cost**: While the paper mentions the additional cost didn't lead to "significant delays in computation", it is not clear why. I believe the paper deserves a more comprehensive discussion about the computational complexity of the proposal. Also, I wonder if the proposed approach becomes prohibitive in some settings.

- **Experiments**: The theoretical analysis does not seem to support the claimed gains on real-world datasets. What are the implications of correctness to top-k diversity/reward? Also, although the paper cites ZINC250K in the Introduction, the experiments only include the QM9 dataset.

- **Technical novelty**: The theoretical contributions of the paper are straightforward. I wonder if the GFlowNet community already knows about the equivalent action problem.

- **Notation**: I found the notation overloaded, which may confuse readers unfamiliar with GFlowNets. For instance, the paper uses the same $P_F$ to refer to the graph-level, state-level policies, and the marginal distribution over terminal states (i.e., $P_F(x)$).

- **Limitations**: The paper does not discuss limitations.

**Questions:**

1. Could the authors provide a detailed analysis of the computational complexity of the proposal? Are there environments where the proposed method becomes prohibitive?

2. Could you provide time comparisons for the real-world experiments?

3. The paper says "A reward exponent of 1 is used for the atom-based task, and a value of 16 is used for the fragment-based task". Was this choice based on prior works? If not, could you elaborate on this choice?

4. Is this the first paper to bring attention to the "action equivalent problem"? Could you elaborate on the impact of your findings on previous works that use GFlowNets for graph generation?

5. I suggest turning Theorem 2 into a Corollary of Theorem 1.

---

> ### Author Response · Authors · 2024-11-19
>
> Thanks for appreciating our work and providing helpful feedback!
>
> # Experiments
>
> ### **Implications of experiments: top-k diversity and reward**
>
> The purpose of measuring top-k diversity and reward is not to demonstrate the correctness of the proposed method. Instead, the experiment aims to provide insights into how unbiased distribution benefits downstream tasks, as diverse and high-reward molecules are crucial for drug discovery. To assess the correctness of our method, we used Pearson correlation.
>
> The effects of bias correction depend on the landscape of the given reward function. If many high-reward molecules are also highly symmetric, the proposed method is more likely to identify these molecules compared to methods without correction. Conversely, if only a few high-reward molecules are symmetric, the correction does not guarantee strong performance in terms of top-k diversity and reward. However, regardless of the task, it is essential to recognize the effects of the correction in advance. Without corrections, there is a risk of inadvertently missing candidate molecules.
>
> ### **Datasets**
>
> To clarify, GFlowNets are trained based on a given reward function and, in principle, do not require a dataset. We referenced the QM9 dataset because the reward function used in our experiments was trained on QM9, which provides important molecular properties, the HOMO-LUMO gap.
>
> The ZINC dataset, on the other hand, was used to construct the fragment vocabulary. However, it was not directly utilized beyond this, as the purpose of GFlowNet is not to learn a distribution from the data but rather to generate graphs guided by the reward function.
>
> We hope this explanation addresses your concern.
>
> ### **Choice of reward exponent**
>
> We found that prior work used wildly different reward exponents $\beta$ for their experiments. For example, the very first GFlowNet paper used $\beta=10$ for sEH experiment, while Multi-objective GFlowNet used 96, and LS-GFN used 6. Our reasoning was that if we use high reward exponent, the training depends more on exploration algorithm and requires longer training, so we chose modest values.
>
> # Technical Contributions
>
> ### **The first paper to study the theoretical foundations on equivalent actions**
>
> To the best of our knowledge, ours is the first paper to study the theoretical foundations of the equivalent action problem, both within the GFlowNet (to our best knowledge, and graph generation communities). While the problem might seem straightforward in hindsight, we would like to emphasize that this recognition often comes after the issue has been formally identified and analyzed. Such seemingly straightforward findings can have a profound impact, as they address fundamental challenges that, once resolved, open new avenues for research and application—something we strongly believe as a strength of our work, not a limitation.
>
> We also highlight that our findings go beyond correcting GFlowNet’s sampling distribution. We introduce a novel method for estimating model likelihood, which has significant implications for a variety of graph-related tasks. This dual contribution demonstrates the broader relevance and potential influence of our work in the field.
>
> ### **Impact of our findings on graph generation**
>
> Although we are not certain how previous works tackled the equivalent action problem, it is highly likely that they employed the approximation $p(s'|s) \approx p(G'|G)$, either intentionally or unintentionally, as evidenced by some open-source GFlowNet implementations. While this approximation leads to an incorrect sampling distribution, we believe that the previous experimental results remain valid, provided the metrics used are consistent and the results are interpreted carefully with the problem in mind. However, we do believe that performance can be improved with the correction we propose, and we recommend that this correction be included in every future work. In addition, we believe our formulation that distinguishes states and graphs is also helpful for clarifying problems in other graph tasks as well; we briefly summarized how our findings can be applied to methods that introduce ‘node ordering’ as a variable in Appendix D.
>
> # Presentation
> ### **Notation**
> We will revise the manuscript and include the table of notations to make it clearer and more user-friendly in the updated version, specifically by distinguishing between states/graphs and transitions/terminal states.
>
> ### **Limitations**
> Thank you for your feedback. We will address the discussion on limitations of our work in the revised version.
>
> ### **Theorem 2 → Corollary**
> Thank you for your suggestion. We are open to modifying Theorem 2 into a Corollary in the revised version, as it is an implication from Theorem 1.

---

> ### Author Response · Authors · 2024-11-19
>
> # Computational Cost
>
> While computing the exact $|\text{Aut}(s)| $ has inherent complexity, as discussed in the paper, this complexity is unavoidable for exact computation. However, irrespective of the computational cost, fixing the sampling bias due to the ignorance of equivalent actions is a fundamental issue that needs to be resolved. **This correction introduces an inherent computational cost, but it is necessary to maintain the consistency of sampling**. In practice, fast heuristic algorithms often perform well, particularly for relatively smaller graphs, and significantly reduce the computational overhead associated with calculating $ |\text{Aut}(s)| $.
>
> We present additional experimental results measuring the compute, as shown below. Note that the scale of the experiments in our paper corresponds to QM9 and ZINC250k.
>
> ---
> | Dataset  | Sample size | Avg. number of atoms (mean ± std) | Compute time for \|Aut(s)\| using _bliss_ (mean ± std) | Compute time for \|Aut(s)\| using _nauty_ (mean ± std) |
> |-|-|-|-|-|
> | QM9| 133885| 8.80 ± 0.51| 0.010 ms ± 0.008  | 0.019 ms ± 0.079|
> | ZINC250k | 249455| 23.15 ± 4.51| 0.022 ms ± 0.010| 0.042 ms ± 0.032|
> | CEP  | 29978| 27.66 ± 3.41| 0.025 ms ± 0.014| 0.050 ms ± 0.076|
> | *Large | 304414| 140.07 ± 49.38|-| 0.483 ms ± 12.600|
> ---
>
> *Large: the largest molecules in PubChem, data retrieved from https://github.com/danielflamshep/genmoltasks. This data is used in the paper “Language models can learn complex molecular distributions.”
>
> **Experiments were conducted on an Apple M1 processor.
>
> ---
>
> When compared to the cost of sampling trajectories, which involves multiple forward passes through a neural network, the compute time for $|\text{Aut}(s)|$ remains still negligible. Also it is important to note that our proposed method requires computing automorphisms only once per trajectory. For comparison, we report the speed of molecular parsing algorithms measured using ZINC250k: 0.06 ms ± 0.70 (SMILES → molecule) and 0.04 ms ± 0.05 (molecule → SMILES). The combination of two parsing steps is often used to check the validity of a given molecule in various prior works. In words, computing $ |\text{Aut}(s)|$ is in an order of magnitude faster than validity checking algorithm.
>
> We used the *bliss* algorithm in our paper. It is easy to use as it is included in the igraph package and is fast enough for our purposes. For large molecules, we can still count automorphisms in few milliseconds using the *nauty* package as can be seen in the table. We observed that the pynauty package does not natively support distinguishing between different edge types, requiring us to transform the input graphs by attaching virtual nodes to handle this limitation. The reported time in the table reflects these preprocessing steps.
>
> While we believe the compute time is already minimal considering current applications, we provide two more recipes to even further improve the run time.
>
> - Data processing tasks can be easily parallelized across multiple CPUs. Since GFlowNet is an off-policy algorithm, $ |\text{Aut}(s)|$ can be computed concurrently with the policy's learning process.
> - For large graphs, fragment-based generation is highly likely to be employed. In such cases, we can utilize an approximate correction formula, as outlined in the paper.
>
> In conclusion, the computational overhead of computing automorphism in practice be minor relative to computation of the entire pipeline.

---

> > ### Comment · Reviewer_ynfw · 2024-11-25
> >
> > I thank the authors for their responses. After reading them, the other reviews, Emmanuel's comment, and the additional comparison to "Baking Symmetry into GFlowNets", I am still leaning towards acceptance and would like to keep my initial score. Importantly, the authors should add the comparison to the revised paper.

---

> > > ### Author Response · Authors · 2024-12-01
> > >
> > > Thank you for your valuable feedback and for recognizing the contributions of our work. We sincerely hope that we have addressed your feedback through the revisions to our paper. In particular, we have included:
> > >
> > > - Comparisons to "Baking Symmetry into GFlowNets" in section 2 and Appendix B.
> > > - “Computational Cost” section on Appendix G.
> > > - New notations to remove notation overloads.
> > > - Limitations in section 7.
> > >
> > > Regards,
> > >
> > > Authors

---

### Public Comment · ~Emmanuel_Bengio1 · 2024-11-20

Dear authors, dear reviewers,

I hope this comment is taken in the constructive spirit it is intended. This paper is a close analog to our NeurIPS 2023 AI for Science Workshop paper, "[Baking Symmetry into GFlowNets](https://arxiv.org/abs/2406.05426)", by George Ma, myself, Yoshua Bengio, & Dinghuai Zhang. In addition, the code provided in the supplementary is, by extrapolating from the appendix and simple inspection, a derivative of an open source `gflownet` repository; of which I am the main contributor and maintainer; and in which features to correct for so-called equivalent actions (idempotent actions in `gflownet`) were [merged into trunk](https://github.com/recursionpharma/gflownet/pull/42) on Jan 19, 2023.

To be clear, the core methodological contribution of this paper is to correct flows by counting automorphisms exactly. In contrast, our paper proposes exactly that, as well as the use of positional encoding matching as an efficient alternative to auto/isomorphism testing.

We understand that this could have happened unintentionally, and we appreciate the effort put into this research, which formalizes and tests this issue more thoroughly than our prior work. That being said, we believe our work provides valuable context and prior art for this submission, and we find this situation disappointing. A simple search using relevant keywords would have easily revealed our paper and the associated code.

Thank you for your understanding.

---

> ### Author Response · Authors · 2024-11-20
>
> Dear Dr. Emmanuel Bengio,
>
> Thank you for bringing this to our attention and for your constructive feedback. We sincerely appreciate the opportunity to clarify and address this oversight.
>
> First, we regret that we were unaware of NeurIPS 2023 AI for Science Workshop paper, *"[Baking Symmetry into GFlowNets](https://arxiv.org/abs/2406.05426).*" Your work on addressing isomorphic (or "equivalent") actions and integrating symmetry considerations into GFlowNets is directly relevant to our research. Had we been aware of your paper, we would have cited and compared it to our work to better position our research within the existing literature.
>
> As you have noted, we utilized the open-source `gflownet` repository, which we referenced in our paper. While we were aware of the option to correct for “idempotent actions,” we found that the implementation enumerates isomorphic actions by performing several isomorphism tests to identify exact isomorphic actions at each transition (get_idempotent_actions function). As noted in the "Baking Symmetry into GFlowNets" paper and in the comments within the function implementation, while this is one of the straightforward methods for correction, it appears to be slow. We question its scalability for real world scenarios. This led us to the mistaken belief (before your comment) that no prior work addressing this issue existed.
>
> Once again, we appreciate your thoughtful feedback and your acknowledgment of our contributions to the additional formalization and testing. In the remaining time, we will update our paper to include comparisons with your work. Thank you for your understanding and for creating GFlowNet.

---

### Author Response · Authors · 2024-11-22

Dear Reviewers,

We provide a detailed comparison with the “Baking Symmetry into GFlowNets” paper.

First, we were pleased to discover that the original authors of GFlowNets had previously recognized this issue and proposed a method to address it—further evidence that this problem is both significant and well-motivated.

---
|  | **“Baking Symmetry into GFlowNets”** | **Ours** |
| --- | --- | --- |
| **Motivation** | Incorporate internal symmetries within the generation process | Train with exact target distribution |
| **Method** | Approximately compute equivalent actions at each transition using positional encodings of a graph. Then, sum their probabilities. | Scale rewards by the order of automorphisms. |
| **Types (Generality)** | Node types  | Node types, edge types, and fragments |
| **Theory** | No theoretical guarantees | Theoretical guarantees on exact learning is provided. The bias without corrections amounts to $\|\text{Aut}(s)\|$. |
| **Experiment** | Synthetic graphs | Both molecules from real-world data and synthetic graphs |
| **Computation** | Multiple positional encoding computations are required at each transition. | Computation of $\|\text{Aut}(s)\|$ is required once for each trajectory. For the approximate method, a summation operation is required over the number of fragments.|
---

The paper “Baking Symmetry into GFlowNets” does not provide sufficient context for full reproduction. However, the open-source `gflownet` code includes the function `get_idempotent_actions` , which we believe serves as an implementation of the paper’s method for exact isomorphism tests. We present the computational cost of performing isomorphism tests using `get_idempotent_actions`, measuring only the compute time of the function call and excluding any preparation code. The QM9 test data (size: 13,389) were used, with trajectories sampled using a uniform backward policy, resulting in an average of 12.72 transitions per trajectory. Although the function must be called for each transition, we report the total cost per trajectory for comparison. Compute times were measured separately for forward and backward actions, and these must be summed to allow a direct comparison with our method.

---
|  | `get_idempotent_actions` (“Baking Symmetry into GFlowNets”) | Our method using *bliss* |
| --- | --- | --- |
| Forward actions | 26.69 ms ± 19.93 | - |
| Backward actions | 4.56 ms ± 5.38 | - |
| Compute time per trajectory | **31.24 ms ± 21.16** | **0.010 ms ± 0.008** |
---

The analysis we provided in our paper for exact isomorphism tests assumes the use of a graph hashing algorithm. The large computational cost of `get_idempotent_actions` arises from its computationally expensive pairwise comparison of actions.

We also observed that the code selectively applies corrections, skipping those for backward equivalent actions when uniform backward policy is used (i.e., when `do_parameterize_p_b` is set to `False`). However, as demonstrated in our paper, equivalent actions should be accounted for regardless of the type of backward policy. We believe that misconceptions on this topic may stem from it being relatively underexplored.

We hope this provides sufficient context for comparisons with prior work. Our method offers an efficient solution to the automorphism problem.

---

> ### Author Response · Authors · 2024-11-24
>
> We provide additional comparisons, focusing specifically on the runtime.
>
> The table below summarizes the runtime for different configurations of atom-based and fragment-based generation methods. To ensure fairness, we re-run all experiments to disable parallel computation, using single processor and GPU. Training steps were limited to 1,000, with all other settings kept consistent with the original paper. We report (mean ± std) with three runs.
>
> ---
> |  | ***Atom** | ****Fragment** |
> | --- | --- | --- |
> | No corrections (Vanilla GFlowNet) | 44.60 min ± 4.69 | 23.84 min ± 0.45 |
> | Exact reward scaling (**ours**) | 49.47 min ± 3.14 | 27.17 min ± 2.62 |
> | Approximate reward scaling (**ours**) | - | 24.92 min ± 2.96 |
> | Isomorphism tests (Ma et al. [1]) | 276.96 min ± 6.28 | 385.12 min ± 12.12 |
> ---
>
> ***Atom:** atom-based generation, with rewards given by a proxy trained on QM9 dataset.
>
> ****Fragment:** fragment-based generation, with rewards given by a proxy that predicts binding energy to the sEH target.
>
> [1] Ma et al., “Baking Symmetry into GFlowNets,” 2024.
>
> ---
>
> When no corrections are applied, the fragment-based method is faster due to its shorter trajectories. However, when exact isomorphism tests are introduced, the computational cost increases significantly. Specifically, the fragment-based method with exact isomorphism tests incurs the highest computational cost (385 min), reflecting the impact of handling larger molecules.
>
> On the other hand, our method introduces minimal additional overhead, making it a practical alternative for both atom-based and fragment-based generation tasks, as the differences in runtime are within the standard deviations. Additionally, we used open-source code for the experiments, making only minor changes to the original implementation. Consequently, there is some additional overhead due to the conversion of data types. We believe this overhead could be eliminated if our method were seamlessly integrated into the pipeline.
>
> We hope this provides additional context regarding prior work and our contributions.

---

### Public Comment · ~Tiago_Silva4 · 2024-11-25
**A question to the authors**

Dear authors,

I recently stumbled upon your paper and, despite being nicely written, I am struggling to understand why Equation (1) is the right way of correcting the automorphism-induced bias in a GFlowNet. To clarify my point, I will assume that the (state-level) reward function is constant, i.e., the target distribution is uniform and that the graphs are unlabeled.

In this case, a naive implementation of a GFlowNet (on the graph-level state graph, $(\mathcal{G}, \mathcal{A})$, with unmodified reward function) would sample each state $s$ in proportion to the size of its equivalence class when balanced, namely, $p_{T}(s) \propto |s|$, instead of uniformly at random. Under the flow network epistemology, the reason for this is that the terminal flow associated to the equivalence class $s$ equals the sum of the terminal flows associated to each of its members. Hence, as the authors noticed, the sampling distribution is inherently biased.

We both disagree, however, in how to eliminate this bias. On the one hand, I would _reduce_ the reward associated to a graph $G$ by a factor of $1/|[G]|$. On the other hand, the authors _increase_ this same reward by a factor of $|\text{Aut}(G)|$ in Equation (1). Nonetheless, in doing so, each state $s$ would be sampled proportionally to $|\text{Aut}(G)| \cdot |s|$, as explained, and the bias would persist. In this regard, it is mostly unclear to me how plugging the equation in Theorem 1 into the TB loss solves the biased distribution problem. Also, it appears to me that the empirical analysis would yield the same conclusions if the scaling factor $|\text{Aut}(G)|$ was replaced by any discrete-valued isomorphism-invariant function.

Additionally, the text contains seemingly conflicting statements, e.g., "If we allow individual graphs to represent states, the equivalence class of a larger graph will be sampled exponentially more often." and "If we do not scale the reward, we are effectively reducing the rewards for highly symmetric graphs". Does the unmodified GFlowNet tends to sample highly symmetric graphs more or less frequently?

Importantly, I may have failed to properly understand this work, and I hope my comments above do not disrupt the review process. Nevertheless, I would be happy to understand the issue with my reasoning. :)

---

> ### Author Response · Authors · 2024-11-26
>
> Thank you for your interest in our paper and for your insightful question!
>
> **Equation (1) is correct, and your reasoning aligns with our findings, assuming that the generation process permits sampling of any graph in $\mathcal{G}$.**
>
> For illustration, let us assume that constant rewards are assigned to terminal states, and we start with a fixed number of isolated nodes in the initial state. In this process, we are only allowed to add new edges. Your reasoning suggests that we should scale the reward by a factor of $1 / |[G]|$. For example, if the terminal graph is “①-②-③”, we should divide the final reward by 3, which corresponds to the number of configurations of different adjacency matrices (that corresponds to “①-②-③”, “②-①-③” and “①-③-②”).
>
> In general, the number of different configurations equals $|[G]| = N!/|\mathrm{Aut}(G)|$, so that $1 /|[G]| = |\mathrm{Aut}(G)|/N!$. Since the initial state has $N$ disconnected nodes, we have $|\mathrm{Aut}(G_0)| = N!$, resulting in $1 /|[G]| = |\mathrm{Aut}(G)|/|\mathrm{Aut}(G_0)|$. This is precisely the scaling term we proposed in the paper!
>
> In practice, however, some graphs are not allowed to be sampled by design. For instance, if we sample graphs node-by-node, the next graph after “①-②” will be either “①-②-③” or “③-①-②”, but there is no way to sample “①-③-②”. To sample “①-③-②”, we would need to allow the policy to sample “③” even from the initial state. This would enlarge the action space and effectively treat node IDs as distinct node types.
>
> In this context, even when using graph-level transitions to model flows, we do not "allow individual graphs to represent states" in such a way that each state is reachable from the initial state. However, when computing graph-level transitions, there is a risk of mistakenly treating graphs as states.  The purpose of the statement was to explicitly distinguish between states and graphs, and it does not imply vanilla GFlowNets were in fact modeling graphs with node IDs.

---

### Author Response · Authors · 2024-11-28

Dear reviewers,

Thank you for your thoughtful and valuable feedback. We have carefully revised our paper to address your comments, aiming to improve clarity and provide a more comprehensive presentation. Notably, we have included thorough comparisons to the workshop paper *“Baking Symmetry into GFlowNets”* to offer readers additional context and to appropriately credit prior work, highlighting how our contributions build upon and extend the existing literature.

We remain confident that our work provides significant contributions to the community by addressing the fundamental issue of equivalent actions in GFlowNets in a rigorous and comprehensive manner and proposing a much more efficient and practical solution.

Below, we summarize the key updates:

- **Comparison with Ma et al. (2024)**: To further position our work in context, we made a detailed comparison to  *“Baking Symmetry into GFlowNets”* paper (Section 2 and Appendix B).
- **Evaluation:** In response to **Reviewer 6mnv**, we included FCS metric (Silva et al., 2024) in our evaluation (Section 6.2).
- **Discussion:** In response to **Reviewer ynfw**, we updated “Discussion and Conclusion” section, which includes limitations as well as implications our work has on prior work in the presence of bias (Section 7).
- **Notation:** We have clarified notations to distinguish between state-level and graph-level policies, eliminating overloaded terms, as suggested by **Reviwer ynfw** (Section 3.2, Section 4.1):
    - $p_\mathcal{S}$, $p_\mathcal{G}$, and $p_\mathcal{S}^{\top}$ now denote the state-level policy, graph-level policy, and marginal state probability, respectively.
    - For backward policy, $q_\mathcal{S}$ and $q_\mathcal{G}$ are used for state-level and graph-level policies, respectively.
    - Forward and backward graph-level actions are now denoted by $\overrightarrow e$ and $\overleftarrow e$, respectively.
- **Experiments:** In response to **Reviewer 6mnv**, we have included additional results using a uniform target distribution to further validate and illustrate our approach (Section 6).
- **Computation:** To **Reviewer ynfw, Rf27**, a new section on computation has been added to the Appendix for completeness (Appendix G).

Overall, we have made several revisions to enhance the paper's readability.

### **Relation to “Baking Symmetry into GFlowNets”**

While the issue of equivalent actions in GFlowNets was identified and partially addressed in the "Baking Symmetry into GFlowNets" paper, the discussion was limited to experimental validation, indicating that equivalent actions could lead to “potentially” incorrect flow probabilities. In contrast, our work provides the first rigorous theoretical foundation for automorphism correction, demonstrating that this issue is not just experimental but a fundamental and systematic challenge tied to graph symmetries, both for atom-based and fragment-based generation.

Now, given the well-supported motivation for addressing this equivalent actions problem, the key question then becomes: **Is there an efficient solution to resolve this fundamental issue** **of equivalent actions in GFlowNets?**

### **Novel Contributions**

In addition to establishing a theoretical foundation, we propose an efficient solution to this problem. Our method applies the correction only once at the end of trajectories, rather than at every transition within a trajectory. This correction involves computing the order of automorphisms, which we found to be computationally efficient even for the largest molecules in the PubChem dataset. The solution is both simple and easy to implement. Any alternative, more "sophisticated" approaches to this problem would, in essence, amount to approximating the order of automorphisms.

Another notable contribution is the introduction of a novel model likelihood estimator that accounts for equivalent actions. Since the estimation of model likelihood is a fundamental measure for all generative models—essential for evaluating model bias, performance, generalization, and more—this contribution has the potential to significantly influence future research.

### **General Implications**

Our findings carry significant implications, especially given that GFlowNets were initially popularized for their reward-matching capabilities. We emphasize that correcting for automorphisms is a fundamental requirement for unbiased sampling, which is critical for applications like molecule discovery. In this regard, our results highlight the need for future work to explicitly detail the methods used to address equivalent actions, ensuring reproducibility and rigorous evaluation. We believe the problem was identified previously, but its significance and seriousness have been emphasized solely in our paper.

We hope this provides additional clarity regarding the contributions of our work, and we would sincerely appreciate any reconsideration of the assigned score in light of these clarifications.

Regards,

Authors

---

### Meta-Review · Area_Chair_RmuF · 2024-12-20

**Metareview:**

**Summary**: This work brings attention to the equivalent action problem with GFlowNets (an important class of generative models, inspired by reinforcement learning, for discrete structured data/graphs). Specifically, the authors analyse this problem theoretically, demonstrating that failing to account for equivalent actions may introduce systematic bias in the sampling distribution for atom-based as well as fragment-based graph generation.  They further relate this bias to the number of symmetries associated with the graph and propose a corrective automorphism-based reward-scaling approach for unbiased sampling, providing empirical validation of its effectiveness.

**Strengths**:  The reviewers appreciated several aspects of this work, notably, the motivation and relevance of the problem, clarity of the presentation, sound theoretical analysis, flexibility of the corrective procedure to accommodate different training schemes  (trajectory balance, detailed balance, and their extensions), and the empirical substantiation.

**Weaknesses**:  The reviewers also raised several concerns and questions pertaining to the additional computational overhead due to the corrective procedure, overloaded notation, lack of discussion on the limitations of the proposed approach, insufficiency of the experiments e.g. in terms of dataset size, fairness of evaluation, metrics not being aligned with goodness-of-fit hindering validation of sampling correctness, and technical novelty.  They also provided several constructive suggestions.

**Recommendation**
Most reviewers provided detailed, insightful reviews. However, one of the reviewers (mRgy) asked only rather generic questions about generative models and did not participate subsequently in the discussions. Therefore, I decided to not consider their evaluation in my recommendation.

I commend the authors for their thoughtful response and additional experiments, which were generally appreciated by the reviewers, prompting many of them to upgrade their assessment of this work.  In contrast to the point raised by one of the reviewers, I do not think the technical machinery needs to be sophisticated/straightforward so long as the theory provides meaningful/actionable/clear insights or interpretations, which this work does.

During the response period, a public comment was posted pointing to a prior work ``Ma, Bengio, Bengio, and Zhang, Baking Symmetry into GFlowNets, NeurIPS 2023 AI for Science Workshop" drawing everyone's attention. This workshop paper had clearly identified the issue of equivalent/idempotent actions in GFlowNet and proposed a closely related method for correcting the flows to address the problem, although without theoretical justification. A part of the code was also leveraged by the authors of the current work.

To their credit, the authors of this work acknowledged their oversight in failing to appropriately position the multiple contributions of that workshop paper (including it being the first to identify the equivalent action issue and providing valuable context for the current work).

At least one reviewer felt that despite the authors' response on this particular issue, the current work loses its novelty - as presented in the original submission - significantly. While I think that the theoretical formalism and the insights provided by the authors here form an important contribution in itself, I cannot disregard this concern about novelty.

Given all the facts and discussion, I believe it would only be fair to the authors of the workshop paper if this paper goes through another review cycle so that the new set of reviewers can make a more informed assessment of the merits of this paper, especially its novelty and repositioning of the contributions with respect to prior work.  I'm therefore not able to recommend a positive decision for this paper at this time. However, I'd like to state that should the program chairs decide to overrule this recommendation, I won't have any (strong) objections.

I hope the authors take this recommendation in the right spirit (though I understand it might be disappointing for them), and use all the feedback and discussion to make a stronger submission.

**Additional Comments On Reviewer Discussion:**

Please see the Metareview for all the details.

---

### Decision · Program_Chairs · 2025-01-22

Reject